# Endocrine Disrupting Chemicals’ Effects in Children: What We Know and What We Need to Learn?

**DOI:** 10.3390/ijms231911899

**Published:** 2022-10-07

**Authors:** Barbara Predieri, Lorenzo Iughetti, Sergio Bernasconi, Maria Elisabeth Street

**Affiliations:** 1Department of Medical and Surgical Sciences of the Mothers, Children and Adults—Pediatric Unit, University of Modena and Reggio Emilia, Largo del Pozzo, 71, 41124 Modena, Italy; 2Microbiome Research Hub, University of Parma, Parco Area delle Scienze, 11/A, 43124 Parma, Italy; 3Department of Medicine and Surgery, Pediatric Unit, University of Parma, Via Gramsci, 14, 43124 Parma, Italy

**Keywords:** endocrine disruptors, exposure, hormone receptor, human health, pregnant women, children

## Abstract

Thousands of natural or manufactured chemicals were defined as endocrine-disrupting chemicals (EDCs) because they can interfere with hormone activity and the endocrine system. We summarize and discuss what we know and what we still need to learn about EDCs’ pathogenic mechanisms of action, as well as the effects of the most common EDCs on endocrine system health in childhood. The MEDLINE database (PubMed) was searched on 13 May 2022, filtering for EDCs, endocrine diseases, and children. EDCs are a group of compounds with high heterogeneity, but usually disrupt the endocrine system by mimicking or interfering with natural hormones or interfering with the body’s hormonal balance through other mechanisms. Individual EDCs were studied in detail, while humans’ “cocktail effect” is still unclear. In utero, early postnatal life, and/or pubertal development are highly susceptible periods to exposure. Human epidemiological studies suggest that EDCs affect prenatal growth, thyroid function, glucose metabolism, obesity, puberty, and fertility through several mechanisms. Further studies are needed to clarify which EDCs can mainly act on epigenetic processes. A better understanding of EDCs’ effects on human health is crucial to developing future regulatory strategies to prevent exposure and ensure the health of children today, in future generations, and in the environment.

## 1. Introduction

Thousands of natural or manufactured chemicals were defined as endocrine-disrupting chemicals (EDCs) because they interfere with hormone activity and endocrine system function, from pregnancy to adulthood [1,2,3,4,5,6,7,8]. These chemicals are world ubiquitous environment contaminants and humans are exposed to several EDCs via the natural flow of air and water, but also by many everyday products including plastic bottles and containers, liners of metal food cans, flame retardants, food, toys, personal care products, detergents, antimicrobial soaps, and household or agricultural pesticides [9,10,11]. Once EDCs enter the food chain, they cannot be easily eliminated from the body because many of them are lipophilic and bio-accumulate in adipose tissue, having a long half-life [12]. Plasticizers do not have these characteristics, so they are usually eliminated from the body in less than 24 h [13,14], but humans are daily exposed to them. Over 800 EDCs cause a great deal of concern because of their “hormone-like action” that can alter cellular, molecular, and epigenetic functions of the endocrine system, both in humans and animals [10,15].

In recent decades, Government Agencies issued several reports and surveys on EDCs’ research state. Guidelines for some EDCs in environmental quality (sediments and water), on tolerable daily intake for foods and beverages, and on fish consumption, were established [16] according to available epidemiological and biomonitoring studies. Current legislation and regulations can be considered ineffective to safeguard humans, nature, and ecosystems. A strategy to prevent EDCs’ exposures will require both political and research projects to limit the use of these chemicals and to develop and implement remediation and bioremediation approaches, also elaborating novel perspectives and sustainable alternative technologies [17].

The incidence and the prevalence of EDCs-associated diseases increased over decades [18]. Epidemiological studies investigating relationships between EDCs’ environmental exposures and human health outcomes provided useful data that must be assessed taking into account the EDCs’ ubiquity, the unpredictable nature of chemical mixtures, the non-monotonic dose–response relationships, and the latent effects. A consequence of the EDCs’ ubiquity is the inexistence of a zero-exposure group, so these chemicals still represent a research challenge because it is difficult to understand the causal link between chemical interactions with biomolecules and outcomes in humans. Most of the published toxicological and epidemiological studies relied on a one-EDC-at-a-time approach to investigate the effects on human health, but this approach fails to consider mixture effects of exposure to many EDCs. It is like a “cocktail effect” towards multiple critical components of human metabolism and in a mixture the different classes of EDCs interact in an either additive or synergistic way, making it more difficult to predict the effects they will provoke and to evince a cause-and-effect association between specific EDCs and endocrine diseases [19]. We know that some EDCs have different hormonal activities. The pesticide dichloro-diphenyl-trichloroethane (p,p′-DDT) acts as an estrogen receptor (ER) agonist, whereas its metabolite dichloro-diphenyl-dichloroethylene (p,p′-DDE) exerts anti-androgenic effect [20]. Bisphenol A (BPA) is an ER agonist, but also antagonizes thyroid hormone receptor (ThR) [21]. Considering the great number of EDCs, it is still unclear how to monitor exposure and how to use or develop assays able to capture the effects of these mixtures. Moreover, it is also unclear how regulatory decisions will be amended to account for exposure to mixtures rather than to single compounds. The prediction for mixture toxicity by proper models would be a supportive tool for the risk assessment and sound regulation of EDCs mixtures in consumer products. To date, although many ingredients in consumer products are potential endocrine disruptors, most of them have not been screened for ER and androgen receptor (AR) transactivation activity (agonistic/antagonistic) [22]. Humans are exposed to low doses of hundreds of chemicals since fetal life, so the development of a multiclass extraction method is of great importance for the identification of these compounds. Moreover, the lifelong environmental exposures (known as the “exposome”) must need to be taken into consideration in risk assessment studies investigating these complex interactions, using methods based on omics technologies [23]. Like endogenous hormones, EDCs are characterized by the non-monotonic dose–response curve, causing maximal physiological effects at low and high doses (“U-shaped”) or at intermediate doses (“inverted U-shaped”) [24,25]. Changes above or below the optimal hormonal level can be detrimental: stronger physiological effects can be seen at lower EDCs doses than at higher doses, so the principle that “the dose makes the poison” is not generally valid for EDCs. This characteristic could be related to differences in tissue sensitivity and response to EDCs or to different receptors in the same tissue that can be activated by different EDCs’ doses [11]. Hence, trying to characterize dose–response functions and to identify safe thresholds by testing a small number of doses may be inefficient for EDCs. It is important to remember that there is a lag-time between the exposure to EDCs and the clinical expression of a disease, so consequences to EDCs’ exposure may not be directly evident but may be manifested many years later [9]. The long-term effects of EDCs’ exposure at different life-times on health and the disease’s risk later in life can be explored through a life-course epidemiology approach. One big challenge is to replace animal testing with alternative test methods (in vitro or in silico) to predict in vivo EDCs’ effect outcomes, but we need to be sure to interpret data appropriately [26].

EDCs can be easily detected in air, water, and dust samples, but ambient monitoring does not reflect the chronic exposure. To date, EDCs complete removal from these environment sources is challenging [27]. Despite biomonitoring does not identify the source of contamination, it is the more appropriate method to detect EDCs in many biological samples (blood, placenta and fetal blood, urine, and adipose tissue), provides data on the individual exposure level at a given time point, and reflects past exposure to persistent EDCs. However, response of each human subject to exposure may be different, ranging from overt toxicity, to subtle dysfunctions, to no effect at all. These differences seem to be due to the genetic background, environmental, occupational and life style factors, diseases, eating habits and source of food consumed [11]. Biomonitoring is invasive and costly and cannot be proposed as a standard routine evaluation, except for occupational exposure. The European Cluster to Improve Identification of Endocrine Disruptors (EURION) is a group of eight research projects, started in early 2019, that aimed to increase the mechanistic understanding of EDCs’ effects and to develop improved methodologies and strategies for chemical testing and hazard assessment [28]. The Organization of Economic Cooperation & Development (OECD) was a first attempt to organize data and knowledge on standardized methods and developed a guide document on tests available for the assessment of EDCs. Adverse effects can be detected in vivo assays such as short-, medium- or long-term rodent bioassays according to OECD test guidelines (levels 4 and 5). However, mechanistic assays, frequently conducted in vitro, are required to obtain information on the potential endocrine mode of action of a substance (i.e., by interference with a hormone receptor or with hormone metabolism, level 2) [29]. However, to predict in vivo outcomes is difficult because of complex EDCs’ toxicokinetic parameters (absorption, distribution, metabolism, and excretion), sensitive windows of exposure, species-to-species differences, gender differences, and quantitative understanding of dose–response relationships.

Our knowledge on EDCs’ numbers and pathogenic mechanisms of action [11,12,15,30,31] and their effects on humans’ health [1,2,3,4,5,6,7,8,32,33] greatly improved over the last decade. Endocrine signals are involved in the control of every development stage and the risk of lifelong EDCs’ adverse health effects is enhanced when periods of exposure coincide with both the formation and the differentiation of organ systems in early development. EDCs can be transferred from mother to child through transplacental route [34,35,36,37] as well as breast milk [38,39]. The timing of exposure to EDCs is very important for the assessment of effects on the endocrine system. In accordance with the Developmental Origin of Health and Diseases (DOHaD) concept [40] in utero, early postnatal life, and/or pubertal development are periods highly susceptible to EDCs’ exposure (Figure 1). Any change in the environment can be “imprinted” in the organism, leading to human health effects and susceptibility to a wide range of diseases and disorders, such as congenital malformations, increased risk for short stature, dysregulation of pubertal development and fecundity, thyroid dysfunction, obesity, diabetes, and metabolic syndrome [2,10,41], but also to the development of infections, asthma, behavioral and learning disorders, cardiovascular disease and cancer in hormonally sensitive tissues later in life [10,11].

Exposure to EDCs represents one of the most critical and ubiquitous public health problem, requiring global solutions. Generally, chemical regulation processes are characterized by three major phases including the collection of hazard (in vivo, in vitro, and in silico analysis) and exposure data, the risk assessment being based on available information to determine whether a management measure is needed to relieve the risk, and the risk management decision to control, reduce, or prevent the potential harm [42]. On these bases, some EDCs are “legacy” contaminants because they were monitored for several decades in environmental matrixes, while others were banned in European Union (EU) and Unites States (US) and limitations were implemented because of clear evidence of adverse effects on the endocrine system. To date, there is still a scientific debate about safe exposure levels to some EDCs that are considered “contaminants of emerging concern” because they are still under research programs. The identification of environmental elements affecting public health has implications for research and represents a great opportunity to improve patient care, prevention, and human health. However, to date, regulatory agencies use different approaches to evaluate the complex data on EDCs, resulting in different end points and methods for gathering and interpreting scientific evidence, arriving at different conclusions for hazard evaluations. Regulatory systems need to be consolidated and simplified to ensure that EDCs are recognized in a timely manner and that actions are effectively taken, minimizing both the human and the environment exposure risks. In EU, EDCs are regulated under the Registration, Evaluation, Authorization and Restriction of Chemicals and general principles that call for the minimization of human exposure, the identification of very high concern substances, and the ban of pesticides use. When scientific evaluations cannot conclude with sufficient certainty, the EU Commission is guided by the “precautionary principle” to take protective measures for citizens and the environment [43,44]. In US, programs are focused on to screen pesticides, chemicals, and environmental contaminants for their potential effect on ER, AR, and ThR, and regulations are strictly risk-based. Minimization of human exposure is unlikely without a clear overarching definition for EDCs and relevant pre-marketing test requirements [45]. An international program is needed to harmonize policies and scientific approaches and to address emerging issues of concern such as the effects of EDCs on human health, the applicability of certain toxicological principles (i.e., the “safe threshold”), the combined EDCs’ exposure, and the development of safer alternatives to substitute known EDCs.

Moreover, education programs by scientists and health professionals should be emphasized as they can be helpful in public awareness to improve the general understanding on EDCs, to educate about behaviors to minimize exposure to potential EDCs and its consequences, especially in early life [10,46], as briefly reported in Table 1.

This review summarizes what we know about EDCs’ pathogenic mechanisms of action and the effects of the main EDCs on endocrine system health during childhood, with scientific and public controversy still surrounding the available data.

## 2. EDCs Mechanisms of Action

EDCs are characterized by high heterogeneity, but they have some key characteristics that allow us to better understand both the mechanisms of action and the consequences on human health.

Despite that EDCs usually display a much lower affinity for hormone receptors compared to natural ligands, they often disrupt the endocrine system by mimicking natural hormones, antagonizing their action, or modifying their synthesis, metabolism, and transport across cell membranes. Moreover, EDCs can induce epigenetic modifications in hormone-producing or hormone-responsive cells [18,47].

Most of the reported harmful effects of EDCs are attributed to their interaction with nuclear receptors functioning as transcriptional regulators in the cell nucleus, such as the members of the nuclear hormone receptor (NHRs) superfamily. NHRs are a class of ligand-activated proteins entering the nucleus and functioning as transcription factors to transactivate or repress gene expression. NHRs without the ligand reside either in the cytoplasm or in the cell nucleus in a complex with chaperones and/or co-repressors (coRe). The activation domain of NHRs binds the hormone inducing conformational changes in the protein that lead to dissociation of the repressive complex, the recruitment of transcriptional co-activators (coAct), and receptor dimerization. NHRs either homodimerize or heterodimerize with retinoid X receptor and this, finally, leads to gene transcription of NHRs target genes [11,31,48,49].

EDCs can inappropriately bind to NHRs acting as agonists or antagonists, enhancing the gene expression or inhibiting the activity of the receptor, causing adverse biological effects. Due to limitations in mechanistic assays, regulation of EDCs over the past decade was focused on substances affecting the so-called “EATS pathway”, in which the main involved NHRs are ER (ERα and ERβ), AR, ThR (ThRα and ThRβ), glucocorticoid (GR), mineralocorticoid (MR), and progesterone (PR) [48,50]. So, EDCs with EATS modalities are primarily associated with reproduction (sex steroid hormones and steroidogenesis) and development/metabolism (thyroid hormone).

Despite most of the studies focusing on ER, AR, ThR and their mechanistic pathways, there are other mechanisms of action involving the aryl hydrocarbon receptor (AhR), a ligand-activated transcription factor which regulates the expression of several genes, including the cytochrome P450 (CYP)-1 gene family members [31,51]. Transcriptional activation of AhR is very similar to NHRs, so it is now considered as an endocrine disruptor target like other EATS receptors [52]. Specifically, the non-activated AhR protein resides in the cytosol bound to a chaperone complex and, upon ligand-mediated activation, the AhR translocates into the cell nucleus forming a heterodimer with the ubiquitously expressed aryl hydrocarbon receptor nuclear translocator (ARNT). The AhR-ARNT complex binds to specific sequences of DNA called xenobiotic responsive element (XRE) in the promoter region of target genes and activates transcription of these genes via the recruitment of transcriptional coAct [53]. Although the AhR was originally discovered as a primary target of the toxic dioxin, it is now recognized that it is able to respond to several EDCs [54].

Figure 2 depicts the main EDC modes of action involving the NHRs and the AhR signaling [31,53]. They can be briefly described as:direct activation of the classical NHRs—EDCs, having similar structure to natural hormones, can enter the cell where some NHRs are kept in an inactive state. Upon EDCs binding, NHRs monomers dimerize and then bind to NHR response elements (NREs), interacting with DNA sequences. The activated dimer can act either as an activator or repressor of gene transcription according to coAct or coRe recruitment to the target gene, respectivelydisturbance of NHRs signaling—EDCs can affect receptor function by interfering with:receptor degradation—NHRs degradation may be regulated by the ubiquitin (Ub)–proteasome pathway that also seems to depend on the AhR. The liganded AhR-ARNT heterodimer can represses the hormone-mediated transcription by targeting NHRs to the Ub-ligase complex, promoting the decrease in some NHRs levels, such as ER and ARcoAct recruitment—the activity of both the NHRs and the AhR depends on transcriptional coAct. Competition between NHRs and AhR for common coAct is a plausible mechanism by which AhR ligands may disturb NHRs signalingDNA-binding competition—the liganded AhR-ARNT heterodimer binds to sequences close to unliganded NHRs binding sites, called inhibitory (i)XREs having slightly different composition than XREs. In this way, the activated AhR can bind but cannot activate gene transcriptiondysregulation of hormone metabolism—enzymes induced by activated AhR are involved in metabolism of xenobiotics but also in the catabolism of steroid hormones. AhR can regulate the levels of circulating estradiol (E2) by controlling the gene expression of CYP involved in estrogen production from cholesterol. Many EDCs can interfere with the enzyme aromatase (CYP19), which converts androgens to 17β-E2 by demethylation and has been shown to be a direct AhR target gene. Thus, activation of AhR by EDCs can lead to increased degradation of steroid hormones and higher E2 production as well.

In summary, EDCs can interfere with the hormonal balance of the body through several different mechanisms. EDCs can modulate hormone receptor expression, internalization, and degradation. The binding of a hormone to a specific receptor triggers intracellular responses that are dependent on the receptor and tissue-specific properties of the target cell. The physiological role of NHRs and AhR as well as their complex mutual crosstalk remain to be determined, as do resulting impacts on human health. With more and more endogenous AhR ligands being discovered, the potential impact of EDCs on hormones’ signaling must be studied in more detail [55].

In recent years, EDCs that disrupt through non-EATS modalities have garnered significant interest from regulators and industry. Non-EATS pathways include retinoic acid, peroxisome proliferator-activated receptors (PPARs), GR signaling, insulin (INS) receptor signaling, gastrointestinal hormones, and cardiovascular-related hormones [56]. The development of new test methods to screen chemicals that can potentially alter these hormone signaling pathways is a challenge.

EDCs can induce epigenetic changes in hormone-producing or responsive cells [57], including the context of transgenerational effects. When EDCs’ exposure takes place during pregnancy and in the early life development, EDCs-induced epigenetic alterations permanently affect the epigenome in the germline, enabling changes to be transmitted to the next generations [58]. As depicts in Figure 1, the exposure (E) of pregnant women to EDCs could involve three generations: the mother (E0), the fetus (E1), and the next generation(s) (E2), not directly through the EDCs’ exposure in utero but through fetus’ germinal cells [59]. Some EDCs are known to interfere with hormone synthesis and to disrupt selective and passive steroid hormones transport across cell membranes. Moreover, EDCs can alter hormone bioavailability by interfering with the distribution of hormones in hormone-responsive tissues or with the circulation of hormones, including the displacing hormones from their serum binding proteins, which can impair the active hormone delivery to target tissues. Finally, EDCs can alter the rates of inactivation of hormones (metabolic degradation or clearance) and the total number or positioning of cells in hormone- producing or responsive tissues.

## 3. EDCs Assessment Methods

The assessment of EDCs in different human biological samples is performed by targeted approaches using both gas chromatography and liquid chromatography mass-spectrometry (MS)-based methods [60,61,62]. Characteristics of the main analytical techniques used for EDCs detection are summarized in Table 2.

Samples treatment plays an important role to achieve reliable results, especially when EDCs’ determinations at low levels are performed in complex matrices, such as biological fluids. The use of sorbent materials able to guarantee a selective EDCs’ extraction is demanded: boosting detection selectivity requires that specific binding interactions dominate over the non-specific ones [63,64,65]. To accomplish this task, molecular receptors must to be devised according to their targets and they need to be structured in architectures which retain the designed molecular recognition functionalities in real-world working conditions. In addition to the powerfulness of molecular recognition strategy, recent advances of MS in terms of sensitivity, resolution, and high-throughput provided effective analytical platforms for untargeted analysis, being high resolution MS a powerful tool for the detection of reliable biomarkers linked to specific diseases [66,67,68]. With epidemiologic evidence on the link between EDCs and endocrine diseases during childhood still limited [1,2,3,4,5,6,7,8,11,33], a deeper investigation exploiting the capabilities of omics sciences to better understand the role of EDCs in exerting health risks is still an open challenge [23,69,70]. Current available omics technologies include methods to measure changes caused by toxicants monitoring global gene expression (transcriptomics), protein (proteomics), metabolism (metabolomics), and microbe (microbiome) levels [42]. A major limitation of epidemiological studies is that they generally measure human exposure to a single EDC or, at best, to a set of isomers or congeners within a family of EDCs. A fuller understanding of potential human health risks requires studying the complex mixtures to which we are exposed. So, to determine the potential toxicological impact of EDCs, an isomer-specific approach is necessary [71,72]. The reliable identification of biomolecules characterized by sufficient sensitivity/specificity to be used in clinical practice is urgently demanded to address important issues of public health concerns, including the implementation of preventive measures addressed mainly to pregnant mothers and childhood.

## 4. The Most Common EDCs

Humans are exposed to EDCs through food production, industrial activity, home and personal care, and medical care. Twelve EDCs causing adverse effects on humans and the ecosystem were initially identified as the “dirty dozen”.

### 4.1. Plastics and Plasticizers

BPA and phthalates are the most studied EDCs with potential effects on human health. BPA and bis-2-ethylhexyl phthalate (DEHP) are considered “non-persistent” chemicals, but they are regularly found in the environment and food due to the continuous release during product production processes. Despite restrictions and prohibitions on their use imposed by legislations, especially in products for children, continuous human exposure to these EDCs may occur through the ingestion of food, inhalation of air and dust in indoor environments, and contact of dust and articles with human mucous membranes (i.e., placing of articles in the mouth) and skin. BPA and phthalates can change the expression of noncoding RNAs, affecting microRNA expression in placental, Sertoli, and breast cancer cell lines [73].

#### 4.1.1. Bisphenols

BPA-based epoxy-phenolic resins are used since the 1950s as protective coatings in beverage and food use cans, as well as a coating for tanks for the drinking water storage. BPA is also used in non-food applications, such as paints, medical devices, surface coatings, printing inks, and flame retardants. Based on the precautionary principle, the Commission Regulation (EU) defined that BPA cannot be used in the manufacture of polycarbonate infant feeding bottles and drinking cups or bottles [74]. BPA was demonstrated to interact with several NHRs, such as the ER (agonist), the orphan receptor human estrogen-related receptor gamma, the AR (antiandrogen), the GR, and the PPARγ, and to interfere with the thyroid axis acting as antagonist of ThR [75]. The strength of BPA to bind ER is much weaker than that of endogenous estrogen; however, the multiplicity of receptors and signaling pathways that may be activated or influenced by BPA may explain estrogenic effects, biological, and health parameters influenced even at very low doses [25]. BPA exposure is associated with increased levels of sex-hormone binding globulin as well as decreased circulating levels of androstenedione and free testosterone [11,76,77]. It can also interact with cell surface membrane receptors, such as the G-protein coupled estrogen receptor [75], and reduces the proteasome-mediated degradation of ERβ [78]. Lastly, BPA was demonstrated to affect pancreatic β-cell through a very rapid closure of ATP-sensitive K+ channels, potentiation of glucose-stimulated Ca^2+^ signals, and release of INS via binding at extranuclear ERβ [79]. It can also act increasing glucose induced INS biosynthesis after binding to extranuclear ERα [80]. In some consumer products, BPA is substituted by bisphenol F (BPF) and bisphenol S (BPS), but a similar or a greater estrogenic activity compared to BPA was reported [81].

#### 4.1.2. Phthalates

Phthalates are di-alkyl or alkyl aryl esters of phthalic acid commonly used as additives to increase the flexibility and other properties of plastic materials. The DEHP is used in the production of toys, upholstery, clothing, adhesives, food packaging, and medical devices. The Commission Regulation (EU) authorized the use of phthalates in plastic food contact material, but in 2018 it was defined that they shall not be placed on the market in toys or childcare articles, individually or in any combination, in a concentration >0.1% by weight of the plasticized material [82]. Considering the migration of phthalates from plastic to food, in 2019 the European Food Safety Authority declared specific tolerable daily intake values for phthalates [83]. Phthalate particles in dust might be a great risk for children, because they touch many things and then put their hands in their mouths. Relationships between exposure to phthalates and non-phthalate plasticizers by house dust, hand wipes, and their corresponding metabolites were evaluated in urine samples of 180 children aged 3–6 years from North America. Dermal contact and hand-to-mouth behaviors were demonstrated to be sources of exposure to these plasticizers and concentrations of monobenzyl phthalate resulted 15-times higher in children who lived in homes with 100% vinyl flooring compared to those who lived in homes without vinyl flooring [84]. DEHP interacts with AR, eliciting anti-androgenic effects [85], which decreases the expression of the MR and inappropriately demethylates MR DNA in testes [86]. It can also interact with the PPARs and the AhR [87]. Despite interactions between phthalates and ThR are not expected due to the high specificity of the ligand binding domain, effects on thyroid signaling were observed, most likely through effects on the thyroid hormone entry into cells or on thyroid hormone distribution [88].

### 4.2. Dioxins and Dioxin-Like Polychlorinated Biphenyls

Dioxins are produced through industrial and incineration processes and can be released into the air during natural processes. More than 90 percent of human exposure is through the food chain (mainly animal products) in which phthalates accumulate because they are absorbed and stored in the fat tissue. The most harmful dioxin is the 2,3,7,8-tetrachlorodibenzo-p-dioxin (TCDD).

Dioxin-like polychlorinated biphenyls (PCBs) were domestically manufactured until 1979, when they were banned by the US Environmental Protection Agency (EPA). Due to their non-flammability, chemical stability, high boiling point and electrical insulating properties, PCBs were used to make electrical equipment and in hydraulic fluids, heat transfer fluids, lubricants, and plasticizers. Humans are exposed by eating or drinking contaminated food, through the breathed air or by skin contact.

Dioxins and PCBs break down very slowly, are extremely resistant to degradation, and emissions released long ago remain in the environment, and therefore are classified as persistent organic pollutants (POPs). Several PCBs are able to bind to the intracellular protein AhR, activating signaling pathways that regulate other nuclear receptors, such as PPARγ or ER involved in adipogenesis [89,90], and can activate human ThR-β-mediated transcription [91].

### 4.3. Pesticides

The main studied pesticides were the p,p′-DDT and its metabolite p,p′-DDE and the hexachlorobenzene (HCB). p,p′-DDT is an insecticide used in agriculture, but it was banned in 1972 in US and in many other industrialized countries due to strong evidences on its adverse health effects and potential endocrine modulatory properties. However, in some countries it is still used to control mosquitoes that spread malaria. Exposure to p,p’-DDT likely occurs by eating foods (meat, fish, and dairy products) or breathing or touching contaminated products. p,p′-DDT and p,p′-DDE persist in body (fatty tissues store) and environment. p,p′-DDT binds to ERα and ERβ [92] and stimulates ER-dependent transcriptional activation and proliferation in different species, including humans. p,p′-DDT binds to the transmembrane domain of the follicle-stimulating hormone (FSH) receptor, a G protein coupled receptor (GPCR), to allosterically enhance its stimulation of cAMP production [93] and it prevents the internalization of the thyroid stimulating hormone (TSH) receptor [94]. p,p′-DDE inhibits androgen binding to the AR and represses androgen-dependent transactivation of the AR [95].

HCB is formed as a byproduct during the manufacture of other chemicals. Widely used as a pesticide until 1965, concerns for its toxicity are due to its persistence in the environment and its potential to bioaccumulate in humans. Exposure to HCB occurs through consuming tainted foods, via inhalation and skin contact.

Lastly, **atrazine,** the most studied chlorinated triazine herbicide, is used to control grasses and broadleaf weeds. It was banned in the EU in 2004 because of its high solubility, mobility in soil, long half-life, and widespread water contamination. Human exposure may be by inhalation and skin absorption during application.

People are exposed to low levels of pesticide residues through their diets because of the widespread use of agricultural chemicals in food production. Nowadays, the health effects of these pesticide residues are still not clearly understood.

Some herbicide apparently increases estrogen synthesis in peripubertal male mice, as evidenced by a decrease in serum levels of testosterone coupled with an increase in circulating levels of estrogen and in the activity of the CYP19 [96]. Moreover, several neonicotinoid pesticides block the JAK–STAT3 pathway to enhance the promoter usage and expression of CYP19 [97].

### 4.4. Perfluoroalkyl and Polyfluoroalkyl Substances

Per- and polyfluoroalkyl substances (PFASs) are widely used in industrial applications, such as firefighting foams and non-stick pan, paper, and textile coatings. Concerns arise because they remain in the environment for an unknown amount of time and may take years to leave the body. People are exposed by the consume of contaminated water or food, the use of products containing PFASs, or the breath of contaminated air.

Perfluorooctane sulfonate (PFOS) and perfluorooctanoic acid (PFOA) are manufactured chemicals with hydrophobic and lipophobic nature, chemical and biological stability. As a result of these properties, they are used in several consumer products and are highly persistent in the environment [98]. Although these two compounds are no longer made in the US, chemical manufacturers replaced them with alternative PFASs, such as GenX. The complexity and the continually changing profile of PFASs production makes it difficult to have an exhaustive characterization of their effects, in addition to the very different metabolism of PFOA between humans and rodents. However, PFOA seems to act on both the nuclear receptor binding to ER and the nonsteroidal receptor as PPARα agonist [11].

### 4.5. Flame Retardants

Brominated flame retardants (BFRs) are used in many consumer and industrial products since the 1970s, to decrease the ability of materials to ignite. These chemicals can contaminate air, water, and soil, therefore, people exposure is through diet and consumer products in every place (home, car, airplane, and work). Novel BFRs can cause endocrine disruption inducing anti-estrogenic activity at certain concentrations and exhibiting GR antagonistic effects. Moreover, they seem to upregulate genes encoding for steroid synthesis enzymes [99].

Polybrominated diphenyl ethers (PBDEs) do not chemically bind with products to which they are added (furniture, electronics, etc.), so they can easily release from these products and enter in air and dust. PBDEs are used in many consumer products as polyurethane foams. People can be exposed by contaminated foods (mainly those with a high fat content), air or dust. Recent findings revealed that prenatal exposure to environmentally relevant levels of PBDE-99 leads to steroidogenesis disorders [100].

### 4.6. Other EDCs

Arsenic is widely distributed in environments and people are exposed to it mainly by drinking water extracted from underground arsenic-contaminated sources, but also through contact with soil, water, and food. Moreover, workers in industries producing this chemical are more exposed [101]. The half-life of inorganic arsenic in humans is about 10 h but, being a naturally occurring element, it is not possible to completely remove it from the environment or food. Arsenic can disrupt the GR system and different cellular processes, leading to the development and progression of several diseases [102].

Glycol ethers are solvents found in paints, cleaning products, brake fluid, and cosmetics. Inhalation and dermal exposures are characteristics and they could be limited by creating household cleaning products and purchasing glycol ethers-free skin care products. The elimination half-life was estimated to be between 66 and 90 h. **Glycol ethers’** exposure was associated with low motile sperm count [103] and higher risk of hypospadias [104]. However, to date, no significant evidence supports that glycol ethers disrupt the endocrine system [105].

The influence of heavy metal on the endocrine system, the possible mechanisms of action, and the health effects were reported [106]. The most studied heavy metals are lead (Pb) and mercury (Hg). Pb was used in a several products (gasoline, paint, plumbing pipes, ceramics, etc.) and its exposure mainly occurs through inhalation of combusted petroleum products and drinking contaminated water. Researches demonstrated that Pb can disrupt the hormone signaling regulating the hypothalamic-pituitary-adrenal axis [107], cause oxidative stress, interfere with intracellular signaling and disrupt glucose homeostasis [108]. A significant negative association between prenatal Pb exposure and birth weight (BW) was demonstrated [109]. Hg can contaminate water, soil, and air entering and accumulating in the food chain. So, diet is the most important way of non-occupational exposure in areas where great amounts of contaminated fish and seafood are consumed. Hg is a known endocrine disruptor that is able to lower progesterone levels, increase estrogen, and adversely affect the steroid synthesis pathway [107]. Prenatal exposure to Hg was consistently associated with lower BW [110].

Lastly, perchlorates are persistent pollutants which environmental sources may be related to natural formation and to human activities (i.e., military purposes, space exploration, fireworks, etc.). People are exposed to perchlorates by transdermal contact, inhalation and ingestion. These compounds were classified as goitrogen due to their ability to inhibit the iodine uptake by thyroid cells disrupting the metabolism of thyroid hormones [111,112]. It is difficult to avoid perchlorate exposure, but its negative effects can be minimized by getting enough iodine in one’s diet or using iodized salt.

Sources and characteristics of chemicals most studied in the last 5 years, because of their disrupting effects on the endocrine system, are briefly summarized in Table 3.

## 5. EDCs and Endocrine Diseases—Evidence in Humans

Incidence and prevalence of several human health problems associated with EDCs increased during the last decades [18]. In this section, we are going to provide an overview of the most recent evidence on EDCs-related endocrine system diseases/disruptions during childhood, focusing on issues mainly related to bisphenols, phthalates, dioxin and PCBs, pesticides, perfluorinated chemicals, and flame retardants.

### 5.1. Pre- and Post-Natal Growth

The placental barrier is not completely impermeable to the passage of harmful substances, so exposure to environmental triggers can affect and permanently reprogram normal physiological responses affecting both the intrauterine and the postnatal life [34]. The insulin-like growth factor (IGF) system represents a critical regulator of growth, especially for fetal development, and EDCs are able to interfere with this system [113]. Moreover, EDCs may play a role in methylation processes as changing the human genome and exposure to EDCs during early life can cause epigenetic shifts that can be transmitted over several generations as “non-transmissible diseases” [114,115].

Correlations between in utero exposure to EDCs and birth outcomes were reported in epidemiological studies, but the results were contradictory. EDCs and mixtures of chemical substances can lead to abortion, preeclampsia, fetal growth restriction and premature birth, and congenital malformations through damage of the placenta [116,117]. However, in humans, both the quantification of chemicals exposure and the timing of absorption during pregnancy are difficult to be evaluated, as a spot biosample is not able to capture exposure to chemicals in a shorter or longer time.

Data on postnatal height growth are still lacking. Exposure to EDCs during pregnancy was demonstrated to be associated with both the BW and the fast weight gain in early childhood, underlining the risk to develop overweightness, obesity [118], and other diseases or dysfunctions later in life. This can be due to the alteration of the “programming” in utero, according to the Barker hypothesis of the “Developmental Origins of Health and Disease” [40].

#### 5.1.1. EDCs Mixtures

Improved understanding of how prenatal exposure to environmental mixtures influences BW is essential to protect child health. Maternal employment during pregnancy was associated with a significant increased risk of at term, low BW newborns and this risk increased with the increasing number of involved EDCs groups, suggesting a possible exposure-response relationship [119]. Conversely, the higher concentrations of POPs measured in newborn dried blood spots, the slightly higher risk for to be large for gestational age (LGA) and to present higher BW [120]. Other data revealed a wide range of in utero exposure scenarios, suggesting that several of the environmental contaminants may be independently associated with fetal growth and affect different aspects of both the pre- and the postnatal growth of the offspring [121,122,123,124]. Prenatal exposure to mixtures of persistent EDCs (PFASs, PCBs, and pesticides) may inversely affect postnatal body size [125].

#### 5.1.2. Bisphenols

BPA can easily cross the placenta and an association with preterm birth was speculated because in the animal models, it was shown that its exposure during gestation reduces placental efficiency with subsequent low BW [126]. Evidence from in vitro and animal studies may provide a better understanding of correlations between environmental BPA exposure and its detrimental effects on child development, despite difficulties to draw direct causal relations of BPA effects on the pathophysiology of childhood diseases/syndromes, due to differences in body system complexity between children and adults, as well as between animal and in vitro models and humans. BPA exposure during pregnancy causes changes in fetal epigenetic programming, resulting in disease onsets during childhood, as well as adulthood [127]. Nowadays, evidence suggested that combined exposure to BPA from dietary and non-dietary sources during pregnancy may contribute to a fetal growth restriction, especially when exposure occurs during the first half [128,129,130]. The fetal growth in humans can also be affected by BPF and BPS, but data are still unclear [131,132,133]. Evidences on BPA influence on pubertal height growth are still lacking, but an inverse association between urine BPA levels and height was observed in boys [134].

#### 5.1.3. Phthalates

Phthalates may be associated with increased odds ratio (OR) of prematurity [116,135] and a recent meta-analysis including a total of 59 studies suggested that maternal exposure to phthalates was related to an increased risk for preterm birth (OR = 1.31) [136]. Possible mechanisms are interference with the placental function via effects on trophoblast differentiation and placental steroidogenesis which could increase the risk for preterm birth. This risk was demonstrated to be magnified in those with certain genetic mutations, highlighting the gene–environment interaction [137]. Moreover, it was suggested that differential DNA methylation may link phthalates exposure in utero to fetal growth having a predictive value for the offspring’s obesity [138]. Gender-specific and trimester-specific effects of DEHP exposure on fetal growth and birth outcomes were demonstrated and were longitudinally confirmed during early childhood growth [139]. Change in auxological data (height, weight, and BMI) in children was reported to be longitudinally induced by phthalates exposure during mother pregnancy [140,141]. However, a recent systematic review and meta-analysis including 22 longitudinal and 17 cross-sectional studies demonstrated that prenatal exposure to DEHP was associated with decreased BMI z-score in children [142].

#### 5.1.4. Pesticides

Prenatal exposure to pesticides was related to increased prematurity and preterm birth and may be differently associated with weight, length, and head circumference at birth [143,144,145]. Conversely, it was found that overall frequencies of exposure to household pesticides had no effects on BW and length. However, small, significant associations between the use of fumigation insecticides and decreased BW and between frequencies of exposure to pyrethroid pesticides and suppression of neonatal length growth were found and need to be confirmed in other studies [146]. Pesticide exposure could be a risk factor for the occurrence of growth disorders in children living in agricultural areas [147]. Published studies suggested that in utero exposure to p,p′-DDT, p,p′-DDE, and HCB, may increase the risk for rapid weight gain in infancy [148,149] and high BMI later in childhood [150,151]. Longitudinal positive associations between prenatal exposure to p,p′-DDE and p,p′-DDT with other obesity-related outcomes were also reported in children [152]. Lastly, the association between in utero POPs exposures and major risk factors for adult cardiometabolic syndrome was recently reported [153].

#### 5.1.5. Perfluoroalkyl and Polyfluoroalkyl Substances

Humans epidemiological findings suggested possible associations between PFASs exposure and fetal and postnatal growth, but data are still controversial. The fetus is exposed to PFASs via active or passive placenta transfer, whereas newborns might be exposed via breastfeeding or the home environment [154]. The relationship between maternal PFASs exposure and adverse pregnancy outcomes was evaluated in a recent systematic review and meta-analysis including 21 relevant studies. Results indicated that maternal exposure to PFOS can increase the risk of preterm birth (OR = 1.20) [155]. Another systematic review evaluated results from 14 studies and demonstrated that in utero exposure to PFOA was associated with decreased mean BW in most studies, but only some results were statistically significant [156]. High exposure to PFASs, especially PFOA, was associated with low BW and small for gestational age (SGA) newborns [157,158,159,160]. However, weak and non-significant associations between PFASs and BW z-scores were also found [161]. Data on relationship between maternal serum PFASs concentrations and longitudinal measures of infant/child anthropometry are not unique [159,161]. In this context, considering discrepancies between studies, the impact of PFASs on health is not yet clear, but it deserves further investigation.

#### 5.1.6. Brominated Flame Retardants

The IGF-1 is important for fetal growth and it was suggested that PBDEs are able to interfere with IGF-1 system secretion [162,163,164]. Epidemiological studies evaluated the correlation between PBDEs and BW, but results were discordant [165,166,167,168,169]. It was also reported that changes in placental DNA methylation might be part of the underlying biological pathway between prenatal PBDEs exposure and adverse fetal growth [170]. Lastly, umbilical cord serum BDE-153 and -154 concentrations were related to reduced adiposity measures at 7 years of age [171].

#### 5.1.7. Arsenic

Chronic exposure to arsenic during pregnancy or early life is still a great global health problem worldwide. Its maternal exposure was associated with decreased BW, length and head circumference, but epidemiologic studies involving large samples are needed to better understand these relationships [172].

Data from studies on pre- and post-natal growth are reported in Table 4.

In conclusion, exposure during critical periods of development, such as fetal and early postnatal life, may have consequences. Fetal growth restriction, premature birth, and low BW have been associated with EDCs exposure, but results from epidemiological studies are still contradictory. Current knowledge on placental and fetal growth, as well as in utero programming of metabolism and endocrine function, highlights the need for the implementation of preventive measures for exposure addressed, in particular to pregnant mothers. Future studies should confirm these preliminary findings and it is also important to evaluate EDCs which have never been studied so far in experimental and epidemiological researches.

### 5.2. Pubertal Development

Pubertal development is characterized by activation of the hypothalamus-pituitary-gonadal and the hypothalamus-pituitary-adrenal axis which are regulated by both inhibitory and stimulatory factors [173,174]. Early puberty occurs in both genders, resulting in a progressive trend towards an earlier age of appearance for first pubertal signs and towards an older age of puberty completion [175] and in recent decades, a progressive shortening of the time of puberty in girls and a consequent increased incidence of precocious puberty and premature thelarche was observed worldwide [176]. Variation of the timing and the progression of puberty is not only genetically dependent, so other key factors are involved in its regulation, and concerns raised on the potential role of EDCs [4,177].

EDCs acting as agonists of ER or antagonists of AR [178] can mimic the physiological effect of estrogens and androgens and can cause hyper-stimulation of hormonal pathways. Moreover, they can bind to intracellular receptors and block the function of endogenous hormones, having anti-estrogenic or anti-androgenic effects [179]. However, in humans it is difficult to provide evidence on causal relationships between EDCs exposure and changes in pubertal timing and we also need to consider the concomitant exposure to low doses of tens or hundreds of chemicals, already occurring from the prenatal age, and the delay between the exposure to EDCs during early childhood and the observation of potential consequences on pubertal timing. Epidemiological researches were mainly conducted in geographical areas where accidental exposure to specific chemicals occurred and suggested that EDCs can determine both the advance and the delay of puberty.

#### 5.2.1. Bisphenol A

The ubiquitous use of BPA results in great exposure to its known estrogenic-like action. Animal studies show that gestational exposure to BPA increased mRNA levels of gonadotropins [180] and neonatal exposure permanently affects gonadotrophin releasing hormone (GnRH) pulsatility and pituitary GnRH signaling [181]. In humans, most of the cross-sectional studies shown that serum and urinary BPA levels were higher in girls with central precocious puberty (CPP) than in controls, suggesting a possible role of BPA in the disease onset [182,183]. Exposure to BPA might be one of the underlying factors of early breast development in prepubertal girls [184] and of alteration in the timing of menarche [185,186]. Data from published studies are conflicting and do not allow us to clearly define if the exposure to BPA plays a role in alterations of the pubertal development timing.

#### 5.2.2. Phthalates

Phthalates act as EDCs, but the mechanisms still need to be better understood. Studies suggested anti-androgenic action and both the agonist and the antagonist action on ER. Exposure to low and high levels of phthalates could disrupt pubertal development in both genders resulting in early or delayed puberty [187], changes in the thelarche [188,189,190], pubarche [191,192], and menarche time [140,193,194]. Although some studies did not find a difference in phthalate metabolites urinary levels between girls with CPP and controls [191,195], in other studies both the plasma and the urinary levels were demonstrated to be significantly higher in girls with CPP compared to those with precocious pseudo-puberty and the healthy ones [196,197]. Lastly, in utero exposure was associated with delayed puberty in normal weight females and with early puberty in overweight/obese males, underlining that body weight may also interfere in such associations [198]. Studies show discrepancies between different phthalates and different analyzed pubertal outcomes and the involvement of hormones and receptors other than androgens should be supposed. It is difficult to carry out human studies and to interpret results.

#### 5.2.3. Dioxins

Dioxins act through AhR and thereby interact with other nuclear receptors. Prenatal exposure to dioxin-like compounds seems to irreversibly affect steroid hormone synthesis. Exposure to dioxins was associated with delayed puberty in boys and delayed thelarche in girls, because of anti-estrogenic effects [199,200]. The slow progression of the breast development towards the adult stage was demonstrated in Belgian girls and it was associated with the high activity of dioxins, while no association was found between exposure to this EDCs and age of menarche onset and pubarche development [201]. In 1976, Seveso (Italy) residents were exposed to high levels of TCDD due to a chemical explosion. Pubertal development was assessed in a cohort of 282 women exposed during post-natal period or childhood, but no change in the age of menarche was demonstrated, despite a 10-fold increase in serum TCDD levels [202].

#### 5.2.4. Pesticides

Data on pesticides derives from evidence of precocious or early puberty in children migrated to Belgium because of international adoptions and previously exposed to the p,p′-DDT in their origin country (i.e., Asia, Africa, and South America) through the transplacental route and during the postnatal period. Mean p,p′-DDE’ concentrations were found to be significantly higher in foreign girls with precocious puberty, both the adopted (*n* = 15/40) and the not adopted (*n* = 11/40), compared to the ones diagnosed with idiopathic or organic precocious puberty which were born in Belgium; in the latter group, only 2 out of 15 patients had detectable concentrations of p,p′-DDT [203]. It was hypothesized that migration can stop the exposure to p,p′-DDT and that precocious puberty can develop indirectly, following the suspension of sex steroids and their environmental analogues negative feedback, and directly, as consequence of accelerated hypothalamic maturation secondary to sex steroids action. Although relationships highlighted in these studies remain theoretical, data on migrant children who developed precocious or early puberty supported the concept that EDCs can influence the pubertal development timing differently, according to the stage of life in which they acted [204]. Epidemiologic data supporting the role of pesticides in pubertal development are limited. The association between p,p′-DDT and p,p′-DDE exposure and the onset of menarche was demonstrated in some studies [205,206] but not in others [207]. Possible mechanisms of action of pesticides include both the anti-androgenic and the estrogenic-like effects and the induction of CYP19. The research project PEACH was recently started, and it will provide much needed information on relationships between the exposure to multiple pesticides and the onset of idiopathic premature thelarche in girls from areas of intensive agriculture practice in the Centre of Italy [208]. Lastly, the peripubertal window was demonstrated to be vulnerable to organochlorine chemicals (dioxins, furans, PCBs, chlorinated pesticides) and lead that can alter the timing of pubertal development also in males [209].

#### 5.2.5. Perfluoroalkyl and Polyfluoroalkyl Substances

Current evidence on the potential impact of prenatal PFASs exposure on long-term reproductive health are still lacking, but data from the Danish National Birth Cohort study suggested gender-specific alterations of the pubertal timing with different prenatal exposure to PFASs [210]. A relationship between the later age of menarche and higher levels of prenatal PFOA exposure was reported [211], but associations between prenatal PFASs exposures and age at menarche were not demonstrated later [212]. These results need to be confirmed because the role of these compounds as complex mixtures remains largely unknown [213].

#### 5.2.6. Brominated Flame Retardants

Exposure to PBDEs during peri-pubertal period seems to be associated with earlier menarche age [214] and premature thelarche in girls [215]. Effects of in utero and postnatal exposure to BFRs on pubertal development were evaluated in children whose mothers were exposed to these EDCs [216,217]. Although published data are contradictory, they suggest that BFRs have estrogenic and androgenic properties and that exposure to these chemicals can have an impact on pubertal development.

Studies also found a possible link between the apparently innocuous topical use of essential oils (i.e., lavender oil present in lotions and creams) and the onset of pre-pubertal male gynecomastia and premature thelarche. Essential oil components could be considered a possible source and a possible link with idiopathic pre-pubertal breast development was suggested, due to their estrogenic and anti-androgenic properties demonstrated in vitro [218,219].

Lastly, attention was also drawn to women who were exposed to EDCs during puberty considering their increased risk to develop breast cancer [220,221]. Epidemiologic and laboratory studies provide evidence that EDCs may reprogram normal progenitor cells in the breast and reprogrammed normal cells are transformed by subsequent hormone exposure [222]. Low-dose EDCs can up-regulate CYP19 mRNA, increase CYP19 activity, significantly increase the CYP19-induced biosynthesis of the breast carcinogen 17β-E2, and increase ERα-positive breast cell proliferation [223].

Data from studies on pubertal development are reported in Table 5.

In conclusion, we have a wide range of epidemiological, experimental, and clinical evidence that lead us to hypothesize a role of EDCs in the dysregulation of the pubertal development, mainly phthalates and BPA. Delayed or precocious onset of puberty were described, but to make clear conclusions is difficult because we need to take into consideration multiple variables, such as type or types of involved EDCs, time and duration of exposure, individual variables, and how fertility will be affected later in life. We need to consider that it is quite impossible for the society as it is, to prevent environmental exposures during puberty both in low-income and high-resource countries. Once again, we underline the need for further research. It will be interesting to follow results from the FREIA research that was funded by the European Commission with the aim to develop new tests for effects of EDCs on female reproduction system [28,224].

### 5.3. Male Reproductive System

EDCs by anti-androgenic and estrogenic effects can affect the male reproductive system. According to the hypothesis of the testicular dysgenesis syndrome, hypospadias and cryptorchidism were related to EDCs that seem to act on both the tubular and the endocrine part of the testes. As a result, testes do not develop regularly, hence becoming at greater risk for cancer and having a lower production of testosterone and other endocrine factors necessary for the normal testicular descent into the scrotum and the normal penile formation [225]. Clinically, it is also important to evaluate the anogenital distance (AGD) which is influenced by prenatal exposure to chemicals with anti-androgenic activity during the critical period of fetal testes development [226].

Intrauterine exposure to BPA during early gestation may adversely affect male reproductive development (congenital or acquired cryptorchidism) [227].

No clear relationship between in utero phthalates exposure and cryptorchidism was found, so further researches are needed [228]. We still need more evidence to confirm the role of prenatal phthalate exposure on hypospadias, but its increased prevalence may be the result of exposure to these EDCs having estrogenic or anti-androgenic properties. Studies were mostly focused on the effects of maternal phthalates exposure [229,230,231]. Phthalates affect AGD [228,232,233], but no association was reported [234].

Dioxins may have estrogenic effects through interaction with the dioxin-AhR/ARNT complex that binds to the XRE on target DNA. Through targeted gene networks, AHR/ARNT regulates or directly intervenes in reproductive system development [235]. No associations between placenta levels of dioxins and congenital cryptorchidism was found [236], while accidental exposure to dioxin via breastfeeding was associated with reductions in sperm concentration, number of motile sperm, and total sperm number [237].

Exposure to OCs pesticides was found to be associated with higher risk for cryptorchidism [238,239]. Data on association between pesticides and hypospadias are few and discordant in results [240,241,242].

Lastly, in China, maternal exposure to PFASs was associated with shorter AGD in boys, providing evidence that they can affect male genital development [243].

Data from studies on male reproductive system are reported in Table 6.

In conclusion, environmental influences on fetal testicular development may result in male reproductive disorders, the prevalence of which has increased. Animal studies implicated phthalates, BPA and parabens, to which humans are ubiquitously exposed. However, results from epidemiological studies are discordant, and limited by the small sample size and/or measurement of chemical exposures outside the most relevant developmental window.

### 5.4. Thyroid Function

Interference with thyroid function has several consequences at all ages, particularly during the development [244]. Thyroid hormones secretion depends on complex mechanisms involving a correct function of the hypothalamus-pituitary-thyroid (HPT) axis and fine mechanisms regulating the combination of iodide with the amino acid tyrosine. EDCs can widely affect this process and thyroid disruption can occur at any level of the HPT axis including thyroid hormones synthesis, release, transport, metabolism, and action on target tissues [2,245,246]. Clinical studies are still few and controversial and three projects within the EURION [28] are focusing on improvement of testing strategies for thyroid hormone EDCs: ATHENA [247], ERGO [248], and SCREENED [249].

In vitro and in vivo studies reported the ability of bisphenols to disrupt thyroid function through multiple mechanisms [250]. Antagonism with ThR affecting the ThR-mediated transcriptional activity, direct action of bisphenols on gene expression at the thyroid and the pituitary levels, competitive binding with thyroid transport proteins, and induction of toxicity in several cell lines are the main mechanisms leading to thyroid dysfunction [251]. The potential developmental toxicity of exposure to bisphenols during pregnancy could affect the thyroid system in the offspring in a gender-specific manner [252]. A systematic review found that many of collected data in children suggested a negative correlation between BPA and TSH levels; however, results were inconclusive with respect to thyroid hormone concentrations and effects on thyroid autoimmunity [253]. The potential impact of BPA in the development of the thyroid gland in children reinforces the advice to limit use of BPA contaminated products. Further prospectively designed studies are needed to better elucidate the association between BPA and mechanisms that can cause thyroid dysfunction and affect organogenesis.

Alterations of thyroid hormone levels in umbilical cord blood and newborns were demonstrated to be related to multiple phthalate metabolites exposure during pregnancy [254,255]. The effects of umbilical cord blood phthalates and prenatal phthalate exposure on thyroid hormones in newborns remain unclear, but the fall of TSH levels in newborns may potentially be delaying their development.

Prenatal exposure to TCDD may alter thyroid function later in life and populations with additional thyroid stress may be particularly susceptible for in utero exposure to thyroid disrupting chemicals [256]. Associations between early life exposure to PCB-153 and p,p′-DDE was demonstrated to impact newborn’s TSH and free thyroxine (FT4) levels [257,258].

Exposure to PFASs can alter circulating thyroid hormone levels. The hypothesis is that there may be combined effects of prenatal exposure to multiple PFAS on maternal and neonatal thyroid function, but the direction and magnitude of these effects may vary across each PFASs [259].

PBDEs structurally resemble thyroid hormones and the thyroid system seems to be sensitive to BDE-47 during pre- and postnatal exposure [260].

Data from studies on thyroid function are reported in Table 7.

In conclusion, EDCs are able to alter the normal thyroid homeostasis. If this occurs in the most critical period of the fetal development, damage to normal psycho-intellectual maturation can occur.

### 5.5. Metabolic Diseases

Metabolic disorders and childhood obesity are of concern and are among the main public health problems worldwide due to their high incidence and negative consequences on health that begin in childhood and clinically manifest in adulthood. The Parma Consensus in 2015 highlighted how EDCs may disrupt metabolic systems during critical periods of development with an impact on non-communicable diseases such as obesity, diabetes and the metabolic syndrome [41]. Some EDCs are well known “obesogens” and “metabolic disruptors” because they can disrupt homeostasis and reward mechanisms and increase individual sensitivity [261,262,263]. Additionally, prenatal exposure, even to low concentrations of EDCs, has an impact on cardiometabolic risk factors in preschool children [264].

PPARγ plays a role in the regulation of adipogenesis [265] and any EDCs acting as agonists on this receptor will directly promote the adipogenesis increasing both the number and the size of fat cells [9,266]. “Obesogens” EDCs contribute to the etiology of obesity in several ways including the promotion of the signal of adipose cell lines to the detriment of other cell lines, the differentiation of pre-adipose tissue towards adipose tissue involving the PPARγ activation, and the promotion of fat deposition and of potential epigenetic mechanisms favoring the activation of the adipogenic genes’ transcription factor. Moreover, many EDCs accumulating in the adipose tissue can promote interactions and changes in the endocrine activity of adipose tissue itself and the homeostatic systems underlying weight control [1,2,41,267,268].

Chemicals killing β-cells or disrupting their function were defined as “diabetogen” and some of them directly cause INS resistance (IR) and defects in INS production and secretion. The “diabetogen hypothesis” suggested that “every EDCs circulating in plasma able to produce IR, independently of its obesogenic potential and its accumulation in adipocytes, may be considered a risk factor for metabolic syndrome and T2D” [269]. Lastly, human studies evaluating EDCs effects on the pathogenesis of type 1 diabetes (T1D) are controversial, but this is one of the fields requiring further studies due to the increasing incidence of T1D worldwide [5].

BPA received large attention as an obesogenic substance and it seems to promote adipogenesis through ER [270,271,272,273]. Acute treatment with BPA causes a temporary hyperinsulinemia, whereas long-term exposure suppresses adiponectin release and gets worse IR, favoring the development of obesity-related syndromes and diabetes. The hyperinsulinemia is attributed to the very rapid closure of ATP-sensitive K+ channels, the potentiation of glucose-stimulated Ca^2+^ signals, and the release of INS via binding at extranuclear ER [79,274]. BPA exposure during the prenatal period was associated with increased blood pressure in girls and blood glucose in boys [275]. Adolescents with polycystic ovary syndrome (PCOS) were found to have significantly higher BPA levels when compared with the healthy ones [276].

Controversial results were found on the association between exposure to phthalates and cardiometabolic risk factors in children and adolescents. A systematic review and meta-analysis including 17 cohorts, 15 cross-sectional, and 3 case–control studies reported that phthalates and their metabolites concentrations were significantly associated with BMI, BMI z-score, waist circumference, and low-density lipoprotein cholesterol, triglyceride, and glycemia. Therefore, prevention of phthalates’ exposure and reduction in their use should be the basis of cardiovascular diseases prevention strategies [277,278].

Among PFASs, PFOA and PFOS exposure increased the risk of cardiovascular diseases more than other types of PFASs [279]. Positive associations were found between maternal serum PFASs concentrations and child overweight/obesity [280]. In T1D, the autoimmune process involving the β-cell could be potentially triggered by PFASs [281]. Prenatal exposure to high PFASs levels can alter lipid profiles in newborns potentially increasing the risk for islet autoimmunity and T1D. Moreover, the interaction between human leukocyte antigens risk genotype and prenatal PFASs exposure was suggested to play a role in altered lipid profiles in newborns at a risk of developing T1D [282].

Data from studies on metabolic diseases are reported in Table 8.

In conclusion, obesity is a multifactorial disease in which altered balance between food intake and physical activity, genetic predisposition, and environmental factors play an important role. Several scientific evidences suggest that exposure to EDCs during sensitive periods, such as prenatal, early infancy, and pubertal times, is able to cause the abnormal distribution of adipose tissue and consequently metabolic complications. As for other health status’, we need more conclusive data on the relationship between EDCs and metabolism. Otherwise, the EURION project [28] includes three other projects focused on integrated approaches for the testing and assessment of metabolism disrupting chemicals: EDCMET [283], GOLIATH [284], and OBERON [69].

## 6. What We Know

Despite difficulties to translate what happens in wildlife to humans and limitations due to conflicting results from published studies related to confounding factors, EDCs’ effects on human health are irrefutable. It is important to emphasize the need for precaution and prevention, educating the public, the media, politicians, and governmental agencies about ways to keep EDCs out of food, water, and air and to protect the human population, mainly developing subjects. However, identification of a direct relationship between EDCs exposure and disease outcomes is complex, because both the exposure to low doses of hundreds of EDCs since in utero and the non-monotonic dose responses were observed.

EDCs represent an emerging global health problem that should not be underestimated, requiring urgent attention and action, especially when dealing with human pregnancy. Pregnant women can be easily exposed to a large number of EDCs by dietary intake or in the workplace. EDCs exposure in vulnerable periods can induce organizational changes, with adverse effects occurring in short and long terms: the years of lag time between exposure and appearance of the disease must be considered to interpret the studies. So, the understanding of EDCs role in neonatal outcomes as well as in the development of endocrine diseases during childhood is still an open challenge. Multiple exposures can result in cumulative effects, a situation that is expected for compounds acting via similar pathways, and also probably for those acting on similar health outcomes via different pathways.

Lots of data have arisen relative to the effects of EDC exposure on growth, puberty, reproductive system, thyroid function, obesity and its metabolic complications, but, we should also underline that findings are often conflicting and methodological limits are present. Many of the reviewed studies present significant limitations, including lack of replication, limited sample sizes, retrospective design, publication biases, and inadequate matching of cases and controls.

Looking to ongoing research, EURION [28] is the result of projects granted from the European Commission’s Horizon 2020 Research and Innovation Programme Call SC1-BHC-27-2018—new testing and screening methods to identify EDCs. EURION is funded for EUR 50 million by the European Commission and each project focuses on the development of the test method able to identify lesser-studied EDCs outcomes, using an “adverse outcome pathway” framework. So, they try to define new tests, including in silico predictive models and high throughput screening, key events in in vitro models, and adverse outcomes in rodents, zebrafish and humans.

## 7. What We Need to Learn

Important issues remain to be resolved.

We need to expand research to emerging “EDCs of interest” and to mixtures of low-dose EDCs. Regulatory measures were taken in EU, US, and member states restricted the use of certain EDCs. Substituents to regulated compounds were used, but there are still uncertainties concerning the safety of this substituents and more generally concerning new compounds or poorly studied compounds (i.e., BPS and BPF as substituents to BPA, the plasticizer 1,2-cyclohexane dicarboxylic acid di-isononyl ester as a substituent to phthalates, and GenX as a substituent to PFOA). We need more information on these and other potential new chemical compound exposures for which we have limited information on sources and use and could not be grouped under known EDCs categories.

EDCs can be assessed in human biological fluids (serum, urine, and breast milk), but their quantification is still difficult. Tests predicting metabolic outcomes useful for the evaluation of the impact of EDCs exposure on health during childhood are lacking. EDCs are not only agonists or antagonists of a single hormone receptor or pathway. New models and tools are needed to better understand how EDCs work: studies of EDCs actions on NHRs need to be extended beyond ER, AR, PR, GR, ThR, and PPARs to other nuclear hormone superfamily members, to membrane steroid hormone receptors, and to enzymes involved in steroidogenesis, hormone metabolism, and protein processing in humans and animal models. Moreover, we need projects with implementation of synergistic omics techniques for biomarker discovery and on the use of cost-efficient methodologies for EDs determination. Besides test development, researchers should better understand the epigenetic effects of EDCs, the effects across generations, and the dose–response functions for EDCs effects in humans. Experimental methods, high throughput omics technologies, epidemiology and human biomonitoring studies, and advanced computational models should be integrated together to provide useful information to regulatory efforts to better characterize suspected EDCs, and their connection to health outcomes. To translate results into the in vivo context, we also need models looking at absorption and distribution in the human body to define whether EDCs reach the target tissue in order to cause an adverse effect.

In this context, we need to perform longitudinal and multigenerational analyses in animals and humans to define the link between exposure to EDCs and health effects that will allow us to identify and develop new intervention and prevention strategies. The greater understanding of the molecular mechanisms underlying childhood diseases will open new frontiers in the development of protocols and guidelines in order to develop future regulatory strategies for primary and secondary prevention of EDCs’ exposure to ensure good health in pregnant women and children today, in future generations, and in the environment. Probably, new additional critical sensitive periods, beyond prenatal and early postnatal, must be tested and design studies should also consider gender, genetic, and population differences in response to EDCs exposures and outcomes.

Lastly, the creation of a biobank storing different kinds of biological samples will open the way to long-term follow-up studies regarding metabolic/endocrine diseases or defects and other diseases.

## 8. Materials and Methods

This review was reported in accordance with the Preferred Reporting Items for Systematic Reviews and Meta-Analyses (PRISMA) guidelines [285]. We retrieved the more recent studies evaluating associations between exposure to the most common EDCs and the endocrine system health in children and adolescents. Search criteria for studies in literature were human studies, full-text papers, article type (clinical trial, randomized controlled trial, review, meta-analysis, and systematic review), language (English), time elapsed from publication date (<5 years), and age (from birth to 18 years). Studies were evaluated as the most significant according to number of cases, multicenter studies, and importance of results for human health. Additional relevant studies were obtained by screening and searching references and papers published before 2017 were cited in the text due to their importance in the review’s aim.

The literature search was performed in the MEDLINE database (accessed through PubMed) until 13 May 2022. The following search terms were used: endocrine disrupting chemicals OR EDCs OR endocrine disrupt *. The MEDLINE search was also performed adding (AND) the terms prenatal growth OR postnatal growth OR growth OR pubert* OR thelarche OR pubarche OR reproductive system OR thyroid OR glucose metabolism OR obesity OR type 1 diabetes to previous search terms. Moreover, we performed a hand-screening of all the reference lists included in papers to discover studies missed in the primary search process. Conference abstracts and qualitative studies (i.e., interviews, letters) were discarded. All authors scored retrieved titles and abstracts independently. Subsequently, full texts of all potentially relevant papers were accurately examined and included only if ECDs exposure levels were examined in relation to auxological, metabolic, and endocrine outcomes. A synthesis of results was provided based on each identified theme.

Overall, 411 papers were initially identified. After the screening of titles and abstracts, a total of 127 articles were discarded, leaving 284 articles to be analyzed. Full-text assessment of these articles was available for all eligible articles. Figure 3 depicts the process of selection of studies.

## 9. Conclusions

EDCs are a global and ubiquitous problem. Although to date we have more information on EDCs’ mechanisms of action and we know the importance of the critical windows of exposure, assessing the impact of human exposure to EDCs is still difficult because adverse effects develop latently and occur at later ages. The overall evidence on a pathogenetic role for EDCs is compelling, but data related to pre- or postnatal exposure are still scarce, so it is difficult to draw definitive conclusions.

In this review, we summarized the most common EDCs having their main adverse effects on the endocrine system in pediatric age. Further studies are needed to better understand which EDCs can act on epigenetic processes. Moreover, a better knowledge of EDCs effects on health is really important to guide future regulatory strategies for prevention of EDCs’ exposure and to ensure a good healthy status in children today, in future generations, and in the environment.

## Figures and Tables

**Figure 1 ijms-23-11899-f001:**
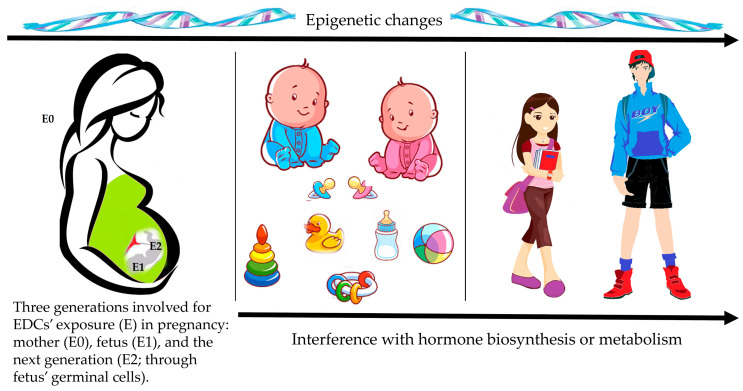
In utero, early postnatal life, and/or pubertal development are periods highly susceptible to EDCs’ exposure, leading to human health effects and susceptibility to a wide range of diseases and disorders through several mechanisms of action.

**Figure 2 ijms-23-11899-f002:**
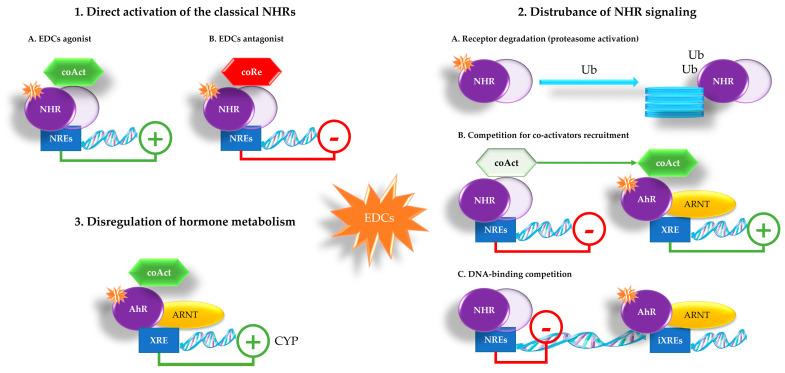
Mechanism of action of EDCs. (**1**) EDCs can directly bind to NHRs acting as (A) agonists inducing the gene expression or as (B) antagonists inhibiting the receptor activity; (**2**) EDCs can affect NHRs function by induction of (A) receptor degradation through proteasome activation, (B) competition for coAct recruitment, and (C) DNA-binding competition; (**3**) EDCs can dysregulate hormone metabolism, mainly inducing degradation of steroid hormones. Abbreviations: AhR, aryl hydrocarbon receptor; ARNT, aryl hydrocarbon receptor nuclear translocator; coAct, co-activators; coRe, co-repressors; CYP, cytochrome P450; EDCs, endocrine-disrupting chemicals; iXRE, inhibitory XRE; NHRs, nuclear hormone receptors; NREs, NHR response elements; Ub, ubiquitin; XRE, xenobiotic responsive element.

**Figure 3 ijms-23-11899-f003:**
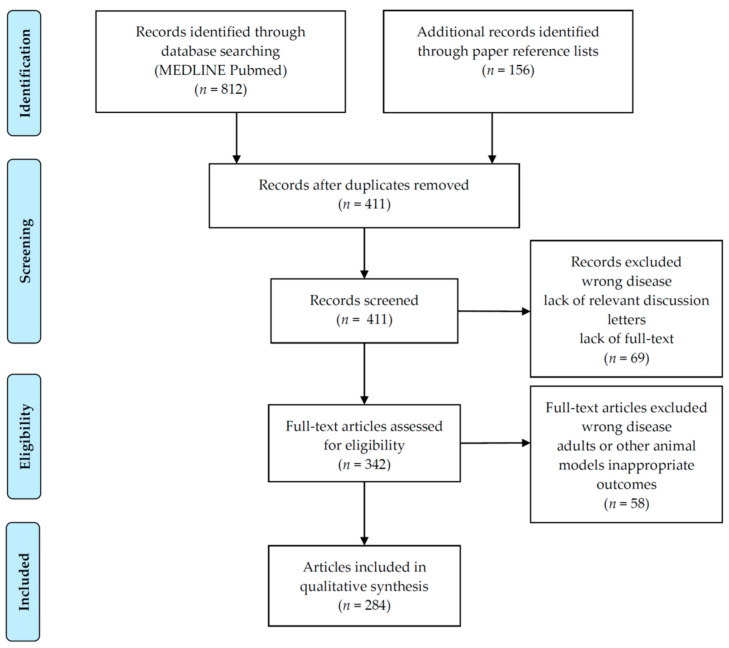
Literature search and studies’ selection: PRISMA flow diagram [285].

**Table 1 ijms-23-11899-t001:** Recommendations towards protection of humans from EDCs [10,46].

It is preferable to opt for: - fresh food instead of processed and canned foods - added chemicals-free food (organic food) - ecological household cleaning products
Avoid plastic containers for food and beverages. Food in plastic containers should not be heated in a microwave oven. Plastic containers can be replaced by glass or ceramic ones.
The consumption of fat dairy or meat products should be reduced.
Avoid handling receipts.
Personal care products should be free of phthalates, parabens, triclosan and other chemicals.
Avoid home cleaning products that are “antibacterial” or carry a fragrance. Regularly clean floors and windowsills to prevent buildup of dust. Indoors environments should be regularly ventilated.
Alternatives to plastic toys are preferred.
Replace or repair furniture that have torn or exposed foam. Flame retardant treated furniture should be avoided

**Table 2 ijms-23-11899-t002:** Characteristics of the main analytical methods for EDCs’ detection.

Analytical Techniques	Advantages	Limitations
Liquid chromatography methods - High-pressure liquid chromatography - Liquid chromatography-high resolution MS - Liquid chromatography-MS	- Selective and reproducible - Small sample amounts - Limited sample preparation - Identification of a multi-class EDCs	- High cost - Require expert analysts - Time consuming - Byproducts
Gas chromatography-MS	- Identification of organic pollutants - Quantification of small amounts in mass concentration - Suitable for biological matrices and environmental screening	- Requires an expert operator - Time consuming - Derivatization step for non-volatile compounds and polar molecules - Interferences into the sample
High-resolution gas chromatography-negative chemical ionization-MS	- Identification of EDCs with quickness, accuracy, and high sensitivity - Identification of compounds with functional groups, such as phenolic compounds - Identification of complex chemical components - Suitable for mixtures	- Complex to use and expensive - Derivatization treatment - Time consuming

**Table 3 ijms-23-11899-t003:** Sources and characteristics of the most studied chemicals causing endocrine disruption.

Food Production		The Most Common EDCs
Agricultural chemicals Food additives Packaging materials	Bisphenols Organochlorines (OCs) PFASs Pesticides Phthalates	BPA: people are mainly exposed through the diet. BPA does not bioaccumulate, so chronic exposure depends on routine exposure to vectors. It is glucuronidated in the liver and excreted in urine within about 48 h. The use of BPA for production of baby bottles was banned in Europe and USA since 2011 and 2012, respectively. It binds with nuclear and membrane-bound ER, AR, GR, as well as ThR and PPARγ Phthalates: people are exposed by eating and drinking foods that have contacted products containing phthalates. Some exposure can occur from breathing phthalate particles in the air. They seem to have a short half-life (<5 h), so their widespread detection is likely due to chronic exposure. They mainly act as anti-androgens (EATS-pathway), but also interacting with PPARs, AhR, and ThR Dioxins and PCBs: people are exposed to dioxin through food chain. Their half-life in the body is estimated to be 7 to 11 years. PCBs’ exposure is by eating or drinking contaminated food, breathing contaminated air or skin contact. PCBs bind to the AhR activating signaling pathways that regulate other receptors (PPARγ or ER). Moreover, they can activate human ThR-β-mediated transcription and can be anti-androgenic Pesticides and HCB: people are exposed to low levels of pesticide residues through their diets. p,p′-DDT and p,p′-DDE have a long half-life (>7 years): they are lipophilic, so they persist in humans because they are stored in the fatty tissues. p,p′-DDT binds to ER, transmembrane domain of the GPCR. p,p′-DDE inhibits androgen binding to the AR and represses androgen-dependent transactivation of the AR. They also seem to induct and/or inhibit the CYP19 PFASs: people are exposed by consuming PFASs-contaminated water or food, using products made with PFASs, or breathing PFASs-contaminated air. PFASs bind to PPARs and to thyroid transport proteins Flame retardants: people can be exposed through diet, consumer products in the home, car, airplane, and workplace. They can bioaccumulate, or build up in people and animals over time, so they remain persistent in the environment for years. *BFRs* seem to induce anti-estrogenic activity at certain concentrations, GR antagonistic effects, and upregulate genes encoding for steroid synthesis enzymes. Moreover, these persistent chemicals can imitate thyroid hormones disrupting their activity
**Industrial Activity**		
Air pollutants Industrial chemical and by-products Water contaminants	Dioxins Fracking fluids Nitrogen oxides Ozone Particulate matter Polyaromatic hydrocarbons (PAHs) PCBs Toxic metals	

**Table 4 ijms-23-11899-t004:** Epidemiological data from some studies on pre- and post-natal growth.

Edcs Exposure	Study Population	Summary of the Main Results	Ref.
** *EDCs mixtures* **			
Maternal occupational exposure to PAHs, PCBs, pesticides, phthalates, organic solvents, BPA, BFRs, metals, and miscellaneous	133,957 mother-child pairs from 11 European cohorts	Higher risk of low BW for exposure to ≥1 EDCs group (OR = 1.25) and to ≥4 EDCs groups (OR = 2.11). The most specific EDCs were pesticides, phthalates, BFRs, and metals	[119]
Newborn dried blood spots concentrations of 11 PCBs, PBDE-47, and p,p′-DDE	2065 newborn, US	Higher concentrations of PCB-52 and PCB-95 were associated with a slightly higher odds of LGA birth (OR = 1.02 and OR = 1.03, respectively)	[120]
Maternal blood samples concentrations for 4 pesticides, 4 PBDEs, 4 PCBs, and 4 PFAS (1st-trimester)	604 pregnant women and their newborns’, US	Higher levels of PBDEs and p,p′-DDE were associated with lower BW and combinations with higher levels of PCBs and PFAS were associated with increased BW	[121]
Maternal serum samples concentrations of six secondary metabolites of phthalates, eight PFASs, PCB-153, and p,p′-DDE	1250 newborns from Greenland, Poland, and Ukraine	MECPP, MeOHP, PFOA, and p,p′-DDE were predictors for lower BW, while exposure to mono(oxo-isononyl) phthalate was associated with higher BW	[122]
Maternal serum samples concentrations of 10 PCB congeners, p,p′-DDT and p,p′-DDE (24th or 36th week of pregnancy)	324 pregnant women and their child, Germany	Significant negative association between the concentration of PCB-183 and length at birth. Concentrations of p,p′-DDE and several PCBs were positively correlated with weight gain in the first 2 years of life	[123]
Cord blood samples concentrations of four PCB congeners, p,p′-DDE, HCB, and heavy metals	1579 mother-newborn pairs, four Flemish birth cohorts	p,p′-DDE and PCB-180 were most consistently associated with BW. An inverse association with BW was found for the PCBs congeners, while an increased BW was observed for elevated levels of p,p′-DDE	[124]
Maternal serum samples concentrations of 8 PFASs, 35 PCBs, and 9 pesticides (median of 15 weeks’ gestation)	425 pregnant mother-daughter pairs, UK	EDCs mixture at the 75th centile compared to the 50th centile was associated with 0.15 lower weight-for-age z-score. At mean EDCs values for 19 months of age, a 0.15 lowering of the weight-for-age z-score corresponds to 0.18 kg lower weight. Weakly inverse associations were also seen for height-for-age and BMI-for-age z-scores	[125]
** *Bisphenols* **			
Maternal blood (8–14 weeks’ gestation and delivery) and umbilical cord blood (delivery) samples concentrations of BPA	80 pregnant women and their newborns’, US	Higher levels of unconjugated BPA both during the 1st-trimester and the end of gestation were associated with a gender-specific reduction in BW and an increase in length of pregnancy	[129]
Maternal urine (3rd-trimester) and neonatal urine samples concentrations of BPA	788 mother-child couples at the 3rd-trimester of pregnancy and 366 mother-child couples during the neonatal period, Korea	BPA exposure was negatively associated with intrauterine linear growth. Maternal urinary BPA levels and birth outcomes were positively correlated	[130]
Maternal urine samples concentrations of BPA, BPF, and BPS (1st-, 2nd-, and 3rd-trimesters)	845 pregnant women and their newborns’, China	Urinary BPS concentrations in the 1st-trimester were significantly associated with reduced BW and ponderal index while concentrations in the 2nd-trimester were significantly associated with reduced BW and birth length. Maternal exposure levels of BPF and BPS for newborns in the 10th percentile of BW and birth length were higher than the ones in the 90th percentile across the period of 10–36 weeks’ gestation	[131]
Maternal urine samples concentrations of BPA, BPF, and BPS	1197 pregnant women and their newborns’, China	Maternal urinary BPA and BPF were negatively related to birth length and positively related to ponderal index. These associations were more pronounced in girls	[132]
Maternal urine samples concentrations of BPA, BPF, and BPS (1st-, 2nd-, and 3rd-trimesters)	1379 pregnant women, Netherlands	Maternal BPS urine concentrations, especially during the 1^st^-trimester, were related with larger fetal head circumference, higher weight, and lower risk of being SGA at birth	[133]
Children spot urine samples concentrations of BPA	754 children, China	Inverse association between urine BPA levels and height was observed in boys. Height z-score at enrolment decreased by 0.49 for the highest BPA exposure levels (90th centile), compared with the lowest ones (25th centile). The inverse association was confirmed considering data 19 months after the enrolment.	[134]
** *Phthalates* **			
Maternal urine samples concentrations of DEHP metabolites (1st, 2nd, and 3rd trimesters)	814 mother-offspring pairs, China	Among males, DEHP levels were negatively related to fetal growth at the 1st trimester, negatively related to both the BW and the birth length at the 2nd trimester, and positively associated with BW at the 3rd trimester. Among females, the 1st-trimester DEHP levels were associated with increased birth length. Data were longitudinally confirmed. Among males, DEHP levels were positively related to 24-month BMI at the 1st trimester, weight gain rates from birth to 24 months at the 2nd trimester, and BMI at 6 and 12 months at the 3rd trimester. Among females, the 2nd-trimester DEHP was negatively associated with BMI at 6 and 12 months	[139]
Maternal urine samples concentrations of 11 phthalates’ metabolites	345 pregnant women and their child, US	Some metabolites resulted strongly associated with BMI z-scores, waist circumference z-scores, and body fat% in children of different ages. Specifically, in the 12-year-old children, in utero levels of DEP, DBP, and DEHP metabolites were positively associated with being overweight or obesity	[140]
Maternal blood samples concentrations of 32 metabolites of 15 phthalates diesters (18–34 weeks’ gestation)	1342 females, Australia	During infancy, a weak negative association was observed between height z-score changes and MHBP, MCiOP, and MEP concentrations. From 2 to 10 years of age, a weak positive relationship between height z-score and higher exposure to MBzP and MECPP was detected. At 20 years of follow-up, associations between phthalates levels and deviation from mid-parental height were not found. Similar results were reported for weight z-score	[141]
** *Pesticides* **			
Estimated agricultural use of methyl bromide near each woman’s for at least one trimester of pregnancy	442 pregnant women and their newborns’, US	Pesticide exposure during the 2nd trimester of pregnancy was negatively associated with weight, length, and head circumference at birth	[142]
Drinking-water exposure to atrazine metabolites and nitrates mixture during 2nd trimester	11,446 pregnant woman-neonate couples, France	Incidence of SGA newborns was increased in exposed mothers. At the 2nd trimester, exposure to 2nd tercile of nitrates without atrazine significantly increased the risk of SGA (OR = 1.74)	[144]
Maternal hair concentrations of 64 pesticides	311 women and their newborns’, France	A significantly higher BW was found for a medium but not a high level of exposure to fipronil sulfone, compared to the lowest exposure level	[145]
History of pesticides exposure in perinatal period, infancy, and childhood	48 children with stunting and 112 controls, Indonesia	Median IGF-1 levels were significantly lower in cases compared to controls. The high level of pesticide exposure and the low IGF-1 levels were significantly associated with stunting	[147]
Cord blood samples concentrations of p,p′-DDT, p,p′-DDE, HCB, and 7 PCBs congeners	379 children, Spain	HCB exposure in the 3rd tertile, compared to the 1st tertile, was associated with higher BMI and WHtR z-score. A continuous increase in HCB levels was associated with higher body fat%, blood pressure z-score (across all ages), cardiometabolic-risk score, and lipid biomarkers (at 14 years). p,p′-DDT exposure was associated with increased cardiometabolic-risk score	[153]
** *Per- and polyfluoroalkyl substances* **			
Maternal and cord blood samples concentrations of PFOS, PFOA, DEHP, and MEHP	29 mother-newborn pairs, Italy	High exposure to PFASs, especially the PFOA, was associated with low BW in newborns	[157]
Maternal blood samples concentrations of PFASs and five OCs	424 mother-child pairs, Norway and Sweden	Prenatal exposure to PFOA, PCB-153 and HCB were associated with higher odds for SGA birth among Swedish mothers. The associations between PFOA and SGA birth were stronger among male offspring	[158]
Newborn dried blood spots concentrations of PFOA, PFOS, PFNA, PFHxS, 3 PBDE, and p,p′-DDE	52 infants with overweight status (at 18 months) and 46 infants with healthy weight status (at 18 months), US	High concentrations of PFOS and PFHxS were associated with lower BW z-score compared to those with low concentrations (more prominent in males than in females). Associations with infant overweight status were not found	[159]
Maternal blood serum samples concentrations of eight PFASs (3–27 weeks’ gestation)	1533 mother–newborn pairs, Sweden	Increased concentrations of PFOS, PFOA, PFNA, PFDA, and PFUnDA were significantly associated with lower BW and BW z-score. Prenatal exposure for PFOS, PFOA, PFNA, and PFDA was also significantly associated with being born SGA. Associations were significant only in girls	[160]
Maternal blood samples concentrations of PFOA, PFOS, PFNA, and PFHxS (16 or 26 weeks’ gestation, or within 48 h from delivery)	345 pregnant women and their child, US	PFASs concentrations, particularly PFOA, were inversely associated with weight and length/height measurements of infant/child from 4 weeks to 2 years of age	[161]
** *Brominated flame retardants* **			
Maternal blood samples concentrations of 10 PBDE congeners (near 26th weeks’ gestation)	286 pregnant women and their newborns, US	Negative associations with BW were seen for BDE-47, BDE-99, and BDE-100. Each 10-fold increase in their concentrations was associated with an approximately 115 g decrease in BW	[165]
Maternal blood (near 12th week gestation) and umbilical cord samples concentrations of 14 PBDEs	686 pregnant women and their newborns, Spain	Inverse associations between BDE-99 and BW and birth head circumference	[166]
Maternal blood samples concentrations of PBDEs and PCBs (mean 12 week gestation)	349 pregnant women and their newborns, Canada	No associations between PBDEs exposure and birth outcomes	[167]
Placental samples concentrations of eight PBDEs congeners	996 pregnant women, China	Prenatal exposure to high level of PBDEs was associated with increased risk of SGA (OR = 2.20)	[168]
Maternal serum samples concentrations of 19 PBDEs congeners (3rd-trimester)	202 maternal-infant pairs (101 with fetal growth restriction cases and 101 controls), China	Concentrations of BDE-207, -208, -209, and the sum of 19 PBDEs were higher in newborns with fetal growth restriction compared with the healthy ones. Increased PBDEs levels were related to decreased placental length, breadth, surface area, BW, birth length, gestational age, and Quetelet index of newborns	[169]

Abbreviations: BFRs, brominated flame retardants; BMI, body mass index; BPA, bisphenol A; BPF, bisphenol F; BPS, bisphenol S; BW, birth weight; DBP, Dibutyl phthalate; DEHP, bis-2-ethylhexyl phthalate; DEP, diethyl phthalate; HCB, hexachlorobenzene; IGF, insulin-like growth factor; LGA, large for gestational age; MBzP, mono-benzyl phthalate; MCiOP, mono-carboxy-iso-octyl phthalate; MECPP, mono(2-ethyl-5-carboxypentyl) phthalate; MEHP, mono-2-ethylhexyl phthalate; MeOHP, mono-2-ethyl-5-oxohexyl; MEP, mono-ethyl phthalate; MHBP mono-(3-hydroxybutyl) phthalate; OCs, organochlorines; p,p′-DDE, dichloro-diphenyl-dichloroethylene; p,p′-DDT, dichloro-diphenyl-trichloroethane; PAHs, polyaromatic hydrocarbons; PBDEs, polybrominated diphenyl ethers; PCBs, dioxin-like polychlorinated biphenyls; PFASs perfluoroalkyl and polyfluoroalkyl substances; PFDA, perfluorodecanoic acid; PFHxS, perfluorohexane sulfonate; PFNA, perfluorononanoate; PFOA, perfluorooctanoic acid; PFOS, perfluorooctane sulfonate; PFUnDA, perfluoroundecanoic acid; SGA, small for gestational age; WHtR, waist-to-height ratio.

**Table 5 ijms-23-11899-t005:** Epidemiological data from some studies on pubertal development.

Edcs Exposure	Study Population	Summary of the Main Results	Ref.
** *Bisphenol A* **			
Urine samples	28 girls with idiopathic CPP and 25 healthy girls, Turkey	BPA levels were significantly higher in the idiopathic CPP group compared with the control group, but they did not correlate with basal serum LH, FSH, and E2 levels	[182]
Urine samples	41 girls with advanced puberty and 47 age-matched controls, Thailand	BPA levels were significantly higher in the advanced puberty group compared with the control group. The median adjust-BPA concentration in girls with advanced puberty who were overweight/obese was greater than in the normal pubertal overweight/obese girls	[183]
Urine samples	25 girls with premature thelarche and 25 healthy age-matched girls, Turkey	BPA levels were significantly higher in girls with premature thelarche compared to the health control group. Weak positive correlations with uterus volume, E2, and LH levels were found	[184]
Urine samples	987 adolescents, US	BPA levels appear to be associated with delayed menarche, particularly for moderate levels of BPA exposure	[185]
Urine samples	655 adolescents, China	Girls with intermediate and high levels of BPA were more likely to have delayed menarche compared to the ones with undetectable levels	[186]
** *Phthalates* **			
Blood samples	41 girls with premature thelarche and 35 healthy controls, US	Higher levels of phthalates were demonstrated in girls with premature thelarche; specifically, measurable values of phthalates were found in 68% of girls with premature thelarche compared to 14% of healthy controls	[189]
Urine samples	29 girls with premature thelarche and 25 healthy age-matched girls, Turkey	MEHP concentrations in girls with premature thelarche were significantly higher (~2 times) than in the control group	[190]
Urine samples	725 girls, Denmark	The highest quartile of urinary phthalates excretion was associated with pubarche delay. No association between phthalates and breast development was found. No difference in urinary phthalate metabolite levels was demonstrated between girls with precocious puberty and controls	[191]
Urine samples	168 children (84 girls), Denmark	Early pubarche was shown in most exposed boys who also had higher testosterone and lower adrenal hormone levels. Pubarche was not altered in most exposed girls	[192]
Urine samples	1051 girls, US	The earlier menarche age, the higher levels of urinary high molecular weight phthalates measured several years before	[193]
Maternal blood serum (18 and 34–36 weeks’ gestation)	369 girls, US	The age at menarche, despite still within the normal range, was slightly delayed in girls exposed at the middle tertile concentration compared to the ones exposed at the lowest tertile	[140]
Urine samples	200 girls, Chile	Higher phthalate concentrations were reported to be associated with earlier menarche among overweight/obese girls	[194]
Maternal urine samples (twice during pregnancy)	338 children (179 girls), US	In utero exposure to phthalates was associated with delayed puberty in females, especially in normal weight ones, and with early puberty in males, especially in overweight/obese ones	[198]
** *Pesticides* **			
Maternal blood samples concentrations of PCBs, p,p′-DDE and other OCs	259 pregnant women and their 213 daughters, US	The earlier was the onset of menarche in girls, the higher was the in utero exposure to p,p′-DDE. The menarche was advanced by about one year for each increase in in utero exposure of 15 g/L	[205]
Serum samples concentrations of p,p′-DDT and its major metabolites	466 women, China	A significant dose–response association between serum p,p′-DDT concentrations and earlier menarche was demonstrated. The mean age at menarche was younger (−1.11 years) in women in the 4th p,p′-DDT quartile compared to the ones in the lowest quartile	[206]
Maternal serum samples concentrations of 9 OCs (pregnancy)	218 girls (case—menarche < 11.5 years) and 230 girls (controls—menarche ≥ 11.5 years), England	No association between in utero exposure to OCs pesticides and early menarche was found	[207]
Blood samples concentrations of 2 pesticides, p,p′-DDE 7 dioxins, 41 PCBs, and 10 furans	516 boys, Russia	PCBs, OCs pesticides, and Pb may delay puberty in boys which pubertal staging and testicular volume were annually evaluated (from 8–9 years until 18–19 years)	[209]
** *Brominated flame retardants* **			
Serum samples concentrations of 6 PBDEs congeners	271 adolescent girls, US	Higher serum PBDEs levels were associated with earlier menarche age	[214]
Serum samples concentrations of PBDEs	124 girls (37 with idiopathic CPP, 56 with premature thelarche, and 31 controls), Italy	Serum PBDEs levels were significantly higher in girls with premature thelarche than in controls	[215]
Maternal serum (near the time of birth) and breast milk samples concentrations of PBBs and PCBs	327 girls, US	Menarche was found to be 1 year earlier in girls who were exposed to high PBBs concentrations in utero and in early infancy through breastfeeding than in girls not exposed or exposed only in utero	[216]
Maternal blood (during pregnancy) and children (9 years old) samples concentrations of 4 PBDEs	623 children (314 girls), US	Prenatal PBDEs concentrations were associated with delayed menarche in girls and early pubarche in boys. No association was demonstrated with breast and pubic hair (in girls) and genitals (boys) development. PBDEs concentrations measured during childhood were not associated with alterations in pubertal timing	[217]

Abbreviations: BPA, bisphenol A; CPP, central precocious puberty; E2, estradiol; FSH, follicle-stimulating hormone; LH, luteinizing hormone; MEHP, mono-2-ethylhexyl phthalate; p,p′-DDE, dichloro-diphenyl-dichloroethylene; p,p′-DDT, dichloro-diphenyl-trichloroethane; Pb, lead; PBBs, polybrominated biphenyl; PBDEs, polybrominated diphenyl ethers; PCBs, dioxin-like polychlorinated biphenyls; OCs, organochlorines.

**Table 6 ijms-23-11899-t006:** Epidemiological data from some studies on male reproductive system.

EDCs Exposure	Study Population	Summary of the Main Results	Ref.
** *Bisphenol A* **			
Maternal serum (10–17 weeks’ gestation)	334 infants, UK	BPA concentrations were positively associated with risk of congenital or acquired cryptorchidism	[227]
** *Phthalates* **			
Interview on maternal occupational exposure during the 1st trimester of pregnancy	471 hypospadias cases and 490 controls, UK	High rate of hypospadias was reported in children of mothers who were exposed to phthalates at work compared with those with no exposure (OR = 3.65)	[229]
Maternal occupational exposure	1202 cases of hypospadias and 2583 controls, Australia	Increased risk of hypospadias in children whose mothers worked in the hairdressing, beauty, or cleaning industry	[230]
Amniotic fluid samples concentrations of DEHP and DiNP (2nd trimester pregnancy)	270 cryptorchidism cases, 75 hypospadias cases, and 300 controls, Denmark	DiNP metabolites were associated with an increased likelihood of hypospadias (OR = 1.69). Concentrations of DEHP were not associated with hypospadias	[231]
Maternal urine samples concentrations of seven phthalates	111 pregnant women, Japan	An inverse association between maternal urine phthalates concentrations and AGD was found in boys, but not in girls	[232]
Maternal urine samples concentrations of 11 phthalate (1st-trimester)	753 pregnant women and their children (380 boys), US	MEHP, MEOHP, and MEHHP concentrations were significantly and inversely associated with measures of boys’ AGD	[233]
Maternal urine samples concentrations of 12 phthalate (28 weeks’ gestation)	245 mother–son pairs, Denmark	No association between phthalates exposure in late pregnancy and AGD in infants at 3 months of age	[234]
** *Dioxins* **			
Placenta samples concentrations of 17 dioxins and 37 PCBs	95 cryptorchidism cases and 185 controls, Finland and Denmark	No association between placenta levels of dioxins or PCBs and congenital cryptorchidism was found at birth and after 3 months	[236]
Maternal serum samples concentrations of TCDD (extrapolation from the concentrations measured soon after the explosion)	39 men (born to mothers exposed to dioxin) and 58 controls, Italy	Men exposed to relatively low dioxin doses in utero and through breastfeeding had a permanently reduced sperm quality (lower sperm concentration, total count, progressive motility, and total motile count)	[237]
** *Pesticides* **			
Maternal breast milk samples concentrations of eight pesticides (1–3 months postpartum)	62 boys with cryptorchidism and 68 controls, Finland and Denmark	Pesticides were found in higher concentrations in 3 months-old boys with cryptorchidism than in controls; no individual compound was significantly correlated with the disease	[238]
Maternal serum and breast milk samples concentrations of seven PCBs, p,p′-DDE, and DBP (3–5 days post-partum)	164 mother-infant pairs (78 cryptorchid and 86 controls), France	Children exposed to high prenatal concentrations of PCBs and possibly also p,p′-DDE have a higher risk for congenital cryptorchidism	[239]
Maternal serum samples concentrations of PCB-153, p,p′-DDE, and HCB (14 weeks gestation)	237 children with hypospadias and 237 controls, Sweden	Increased risk of hypospadias among children from women with p,p′-DDE concentrations at the highest quartile (compared to those in the 1st quartile) and with the highest exposure quartile of HCB (compared to the three lowest quartile) (OR = 1.65)	[240]
Maternal serum (3rd trimester) samples concentrations of p,p′-DDE and 11 PCBs	217 children with cryptorchidism, 197 with hypospadias, and 557 controls, US	No association was found between EDCs’ levels and cryptorchidism/hypospadias	[241]
Questionnaire to estimate total maternal consumption of atrazine via drinking water (6–16 post-conception)	343 children with hypospadias and 1422 controls, US	A weak association between hypospadias and maternal consumption of atrazine was found	[242]

Abbreviations: AGD, anogenital distance; BPA, bisphenol A; DBP, dibutyl phthalate; DEHP, bis-2-ethylhexyl phthalate; DiNP, di-isononyl phthalate; HCB, hexachlorobenzene; MEHHP, mono-2-ethyl-5-hydroxyhexyl; MEHP, mono-2-ethylhexyl phthalate; MEOHP, mono-2-ethyl-5-oxohexy; p,p′-DDE, dichloro-diphenyl-dichloroethylene; PCBs, dioxin-like polychlorinated biphenyls; TCDD, 2,3,7,8-tetrachlorodibenzo-p-dioxin.

**Table 7 ijms-23-11899-t007:** Epidemiological data from some studies on thyroid function.

EDCs and Sample	Study Population	Summary of the Main Results	Ref.
** *Bisphenols* **			
Maternal urine samples concentrations of bisphenols (<18, 18–25, >25 weeks’ gestation)	1267 pregnant women, 853 newborns and 882 children, Netherlands	Higher late pregnancy maternal BPA levels were associated with higher TSH levels in female newborns and higher FT4 levels in males during childhood	[252]
** *Phthalate* **			
Maternal urine samples concentrations of nine phthalate (16 and 26 weeks’ gestation)	389 pregnant women and their newborns, US	Alterations of thyroid hormone levels were demonstrated in both the maternal serum and the umbilical cord blood. Specifically, at 16 weeks of pregnancy for each 10-fold increase in maternal urinary MEP, maternal serum total thyroxine decreased by 0.52 μg/dL. At 16- and 26-weeks’ gestation, for each 10-fold increase in average of maternal urinary MBzP, cord serum TSH decreased by 19%	[254]
Maternal urine (3rd-trimester) and cord blood samples concentrations of five phthalate	61 pregnant women and their newborns, Taiwan	High MBP levels in umbilical cord blood levels were significantly and negatively associated with cord serum TSH and FT4	[255]
** *Dioxins* **			
Prenatal TCDD exposure as: (1) maternal initial TCDD concentration measured in serum samples collected soon after exposure and (2) maternal TCDD estimated at pregnancy	570 children (288 girls), Italy	Compared to the lowest quartile, maternal serum TCDD concentrations at higher quartiles were associated with lower FT3. Children with high thyroid antibodies had significantly inverse associations between maternal serum TCDD and both the TSH and the FT3 levels were stronger than in subjects with normal antibody status. Similar results were found for TCDD estimated at pregnancy	[256]
** *Pesticides* **			
Cord blood and breast milk (14 days after delivery) samples concentrations of PCB-153, p,p′-DDE, and HCB	1784 mother–child pairs, Belgium, Norway, and Slovakia	Early-life exposure to PCB-153 and p,p′-DDE was demonstrate to impact newborns’ TSH levels. Newborns in the 3rd-exposure quartile had TSH lower by 12–15%	[257]
Maternal blood samples concentrations of PCBs (3rd trimester)	497 mother-newborn pairs, Japan	Exposure to PCBs during pregnancy was demonstrated to increase maternal and neonatal FT4 levels	[258]
** *Per- and polyfluoroalkyl substances* **			
Maternal plasma samples concentrations of six PFASs (early pregnancy)	726 pregnant women and 465 neonates, US	In infants, higher concentrations of PFHxS were associated with lower total thyroxine levels, mainly in males	[259]
** *Brominated flame retardants* **			
Cord blood and blood (age 2–3 years) samples concentrations of PBDEs	158 children, US	Children with high exposure to BDE-47 during the prenatal period or toddler age had significantly lower mean TSH levels compared to the ones with low exposure throughout early life. Associations with postnatal exposure may be stronger among boys compared to girls	[260]

Abbreviations: BPA, bisphenol A; FT3, free triiodothyronine; FT4, free thyroxine; HCB, hexachlorobenzene; MBP, mono-butyl phthalate; MBzP, mono-benzyl phthalate; MEP, mono-ethyl phthalate; p,p’-DDE, dichloro-diphenyl-dichloroethylene; PBDEs, polybrominated diphenyl ethers; PCBs, dioxin-like polychlorinated biphenyls; PFASs perfluoroalkyl and polyfluoroalkyl substances; PFHxS, perfluorohexane sulfonate; TCDD, 2,3,7,8-tetrachlorodibenzo-p-dioxin; TSH, thyroid stimulating hormone.

**Table 8 ijms-23-11899-t008:** Epidemiological data from some studies on metabolic diseases.

EDCs and Sample	Study Population	Summary of the Main Results	Ref.
** *Bisphenols* **			
Maternal urine samples concentrations of BPA (6.3–15 weeks’ gestation)	719 mother–child pairs, Canada	Higher levels of BPA concentrations were associated with increased central adiposity outcomes among girls during early childhood	[271]
Urine samples concentrations of BPA	210 boys, Spain	BPA concentrations were associated with increased BMI z-scores and a higher risk of overweight/obesity. Children with higher urinary BPA concentrations also had higher WHtR values and a greater risk of abdominal obesity	[272]
Food concentrations of BPA and BPS and estimation of dietary exposure	585 adolescents (46.6% girls), Spain	Girls <14 years old compared to the older ones had a greater risk for high dietary exposure (3rd tercile) to bisphenols overall (OR = 4.77), as well as BPS (OR = 4.24). Overweight/obese girls were at a greater risk of having high dietary exposure to total bisphenols (OR = 2.81) and BPA (OR = 3.38) than normal weight ones. The more sedentary boys had a greater risk of being included in the 3rd-tercile with regard to bisphenols dietary exposure (OR = 1.10)	[273]
Maternal urine sample concentrations of BPA	218 pregnant women and their children 2 years old, China	BPA exposure during prenatal period was associated with increased blood pressure in girls and blood glucose in boys	[275]
Urine sample concentrations of BPA	62 girls with PCOS and 33 controls, Turkey	Adolescents with PCOS had significantly increased levels of BPA when compared with the control group. BPA levels were significantly correlated with polycystic morphology on ultrasound but not with obesity androgen levels, or other metabolic parameters	[276]
** *Phthalates* **			
Maternal urine samples concentrations of 11 phthalate and 9 phenols (14.0 and 26.9 weeks’ gestation)	309 pregnant women and their children 5 years old, US	Prenatal urinary concentrations of MEP, MCNP, and cumulative mixture were associated with an increased BMI z-score and overweight/obesity status at age 5	[278]
** *Per- and polyfluoroalkyl substances* **			
Maternal serum samples concentrations of two PFASs and five OCs (17–20 weeks’ gestation)	412 pregnant women and their children with overweight/obesity, Norway and Sweden	In children at 5-year follow-up, both the BMI z-score and the triceps skinfold z-score were found to be increased per logarithmic-unit increase in maternal serum PFOS concentrations. Increased odds for child overweight/obesity for each logarithmic-unit increase in maternal serum PFOS and PFOA levels were found	[280]
Serum samples concentrations of PFOS and PFOA	25 children at the onset of T1D and 19 controls, Italy	PFOS levels were significantly higher in patients at the onset of T1D compared to healthy controls	[281]

Abbreviations: BMI, body mass index; BPA, bisphenol A; BPS, bisphenol S; MCNP, mono-carboxy-isononly phthalate; MEP, mono-ethyl phthalate; MHBP mono-(3-hydroxybutyl) phthalate; OCs, organochlorines; PCOS, polycystic ovary syndrome; PFASs perfluoroalkyl and polyfluoroalkyl substances; PFOA, perfluorooctanoic acid; PFOS, perfluorooctane sulfonate; T1D, type 1 diabetes; WHtR, waist-to-height ratio.

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
