# Peer review of "Endocrine Disrupting Chemicals’ Effects in Children: What We Know and What We Need to Learn?"

_ijms, 2022, doi:10.3390/ijms231911899_

Round 1

Reviewer 1 Report (Previous Reviewer 3)

The text has been improved, including some corrections and notes. The EDCs topic are challengable. The present article shows the MOAs, risk assessent evaluation, Public Health aspects and tatical regulatory controls.

Reviewer 2 Report (Previous Reviewer 2)

In the revised version of the manuscript, the authors have adequately addressed my comments and suggestions. As a result, I do not have any further comments and recommend this manuscript be accepted.

This manuscript is a resubmission of an earlier submission. The following is a list of the peer review reports and author responses from that submission.

Round 1

Reviewer 1 Report

In this manuscript, the authors summarize the EDCs’ pathogenic mechanisms of action, as well as effects of the most common EDCs on endocrine system health during pediatric age. In addition, the authors also discuss and point out that further studies are needed to clarify which EDCs can mainly act on epigenetic processes and a better knowledge of EDCs effects on health is crucial for future regulatory strategy for prevention of EDCs’ exposure to ensure a good health in children today, in future generations, and in the environment. In my opinion, this manuscript deserves to be recommended for publication in International Journal of Molecular Sciences. However, there are some minor issues need to be address before this article to be published.

  1. It is better to add outline in the manuscript so that readers can understand it quickly.
  2. The title of each chapter (i.e. results, discussion) doesn’t generalize the content.
  3. A table can be included in the 2.2 to summarize the most common EDCs.
  4. There are some papers talking about the Endocrine Disrupting Chemicals. For example, the paper named “Endocrine-disrupting chemicals and child health” also summarized EDC-caused effects. What dose the author think about the highlight of this manuscript?

Reviewer 2 Report

Detailed comments:

-What was the novelty of this review?

-Line 13: Some readers may find man-made dated or non-inclusive in this context. Consider using a gender-neutral term.
Such as manufactured or artificial.

-Lines 20-21: Rewrite as: while humans' "cocktail effect" is still unclear.

-Line 22: Rewrite as: Human epidemiological studies suggest that EDCs affect.

-Lines 25-27: Rewrite this sentence as:
A better understanding of EDCs' effects on human health is crucial to developing future regulatory strategies to prevent exposure and ensure the health of children today, future generations, and the environment.

-Introduction is very brief! You may bring more explanation dealing with the problems ans solutions by using recent references.

-Figure 1 quality is low. Replace with more quality version.

-Lines 73-75: Rewrite as:
This review summarizes what we know about EDCs' pathogenic mechanisms of action and their effects on the endocrine system health during childhood, with scientific and public controversy still surrounding the available data.

-Results: The mechanisms of action section are not well structured. Please do that.

-Line 77: The mechanisms of action related with EDCs presented in this review is not complete!!!
There are so many mechanisms of action and the authors just presented few ones. You may complete this important section. Check this published paper for seeing these mechanisms of action:
https://jme.bioscientifica.com/view/journals/jme/43/1/1.xml 

-Figure 2: This figure is not acceptable! because it is not involved all mechanisms of action related to EDCs. Revise and redesigned this figure.

-Line 148: As you may know, the most common EDCs are including these 12:
BPA, Dioxin, Atrazine, Phthalates, Perchlorate, Fire retardants, Lead, Arsenic, Mercury, Perfluorinated chemicals (PFCs), Organophosphate pesticides, Glycol Ethers. So, complete your data in this section.

-Line 293: Associated disorders and diseases with EDCs presented here is not comprehensive and complete!!! You may use this published paper to complete this section: https://www.ncbi.nlm.nih.gov/pmc/articles/PMC3443608/

-Line 294: You could use a table instead of this long paragraph information.

-Pages 7-20: there are so extended data and information dealing with EDCs disorders. I suggest to summarize this section and present some tables instead of these Additional explanations.

Reviewer 3 Report

The papers refers to EDC compounds, mode of action (MoA) and feedback relations.

It´s a challenge topic, requering high level of analytic performance in monitoring, as isomers for example.

The regulatory inventary is another task, Agencies have been dedicated to perform protective measures or standards.